# SARS-CoV-2 rebound and post-acute mortality and hospitalization among patients admitted with COVID-19: cohort study

Ka Chun Chong [1], Yuchen Wei[1], Katherine Min Jia[2], Christopher Boyer [2], Guozhang Lin[1], Huwen Wang[1], Conglu Li[1], Chi Tim Hung [1], Xiaoting Jiang[1], Carrie Ho Kwan Yam[1], Tsz Yu Chow[1], Yawen Wang[3], Shi Zhao [1,4], Kehang Li [1], Aimin Yang[5], Chris Ka Pun Mok [6], David SC Hui [7], Eng Kiong Yeoh [1] & Zihao Guo [1]

Recent investigations have demonstrated a relationship between the persistence of SARS-CoV-2 and post-COVID-19 conditions. Building upon a potential connection between SARS-CoV-2 persistence and early virologic rebound, we examine the association of early virologic rebound with post-acute mortality and hospitalization due to post-acute sequelae among hospitalized patients with COVID-19 in Hong Kong. Our study includes 13,859, 3959, and 4502 patients in the all-patient, nirmatrelvir/ritonavir, and molnupiravir group, respectively. Results show that patients who experienced virologic rebound exhibited a significantly higher risk of post-acute mortality (hazard ratio [HR], 1.52; 95% confidence interval [CI], 1.36–1.70) with a risk difference [RD] of 7.19%, compared with patients without virologic rebound. A similar increase in the risk of post-acute mortality is also observed in nirmatrelvir/ritonavir-treated patients (HR, 1.78; 95% CI, 1.41–2.25; RD, 12.55%) and molnupiravir-treated patients (HR, 1.47; 95% CI, 1.18–1.82; RD, 4.90%). The virologic rebound may thus serve as an early marker for post-COVID-19 condition, enabling healthcare officials to monitor and provide timely intervention for long COVID.

The ongoing emergence of new variants of the SARS-CoV-2 virus, exhibiting varying levels of transmissibility and severity, continues to cause substantial mortality and morbidity observed due to coronavirus disease 2019 (COVID-19). Post-COVID-19 conditions, also known as long COVID, post-acute COVID-19 sequelae, or post-COVID-19 syndrome, encompass a range of long-term persistent health burdens that individuals experience after recovering from the acute phase of COVID-19. According to a systematic review[1], at least 45% of cases experienced at least one unresolved symptom after an acute SARS-CoV-2 infection.

SARS-CoV-2 rebound typically refers to a recurrence of signs or symptoms or a return to SARS-CoV-2 test positivity after an initial

---

[1]School of Public Health and Primary Care, The Chinese University of Hong Kong, Hong Kong, China. [2]Center for Communicable Disease Dynamics, Department of Epidemiology, Harvard T.H. Chan School of Public Health, Boston, MA, USA. [3]Division of Landscape Architecture, Department of Architecture, Faculty of Architecture, The University of Hong Kong, Hong Kong, China. [4]School of Public Health, Tianjin Medical University, Tianjin, China. [5]Department of Medicine & Therapeutics, Faculty of Medicine, The Chinese University of Hong Kong, Hong Kong, China. [6]Li Ka Shing Institute of Health Sciences, Chinese University of Hong Kong, Hong Kong, China. [7]S.H. Ho Research Centre for Infectious Diseases, Chinese University of Hong Kong, Hong Kong, China. ✉e-mail: yeoh_ek@cuhk.edu.hk; zihaoguo@cuhk.edu.hk

recovery from COVID-19[2]. While virologic rebound unlikely represents reinfection or resistance to antiviral treatment, patients experiencing virologic rebound may still be infectious for transmitting the virus to others. In the absence of antiviral treatment, approximately one-third of individuals experienced virologic rebound, though only 3% of patients had both symptoms and virologic rebound[3]. In those receiving nirmatrelvir/ritonavir or molnupiravir, the incidence of virologic rebound is similar to those not receiving antivirals and is often accompanied by symptom rebound[2,4–7]. Nevertheless, inconsistent findings were reported in several studies suggesting more frequent virologic rebound in treated individuals e.g.,[8,9].

Recent investigations by Zuo et al.[10] and Ghafari et al.[11] have demonstrated a relationship between the persistence of SARS-CoV-2 and post-COVID-19 conditions. Building upon these studies, along with the potential connection between SARS-CoV-2 persistence and early virologic rebound, we hypothesize that there is a relationship between SARS-CoV-2 rebound and post-COVID-19 conditions. In this study, we examined the association of virologic rebound with post-acute mortality and hospitalization due to post-acute sequelae among hospitalized patients with COVID-19 in Hong Kong. To examine the effect modification of antivirals, our study also assessed the association of interest in patients treated with nirmatrelvir/ritonavir and molnupiravir.

## Results

A total of 88,643 patients with COVID-19 were screened in this study (Fig. 1). Among the 13,859, 3959, and 4502 patients included in the all-patient, nirmatrelvir/ritonavir, and molnupiravir group, 1573 (11.4%), 417 (10.5%), and 559 (12.4%) experienced virologic rebound, respectively. The median (interquartile range) follow-up time for the post-acute death were 365 days (309–365), 365 days (325–365), and 365 days (315–365), respectively.

In the all-patient group, 914 (58.1%) were males and the mean (standard deviation) age was 74.6 (15.8) years among those with virologic rebound, whereas 6451 (52.4%) were males and the mean (SD) age was 76.5 (15.6) years among those without virologic rebound (Table 1). A similar age and sex distribution was observed in the nirmatrelvir/ritonavir and molnupiravir groups. The cumulative incidence curves of post-acute mortality by the three patient groups without adjustment for baseline covariates are displayed in Fig. 2. After

Standardized mortality ratio weighting (SMR), all baseline covariates were well-balanced with absolute standardized mean differences (SMDs) <0.1 across different patient groups.

## Post-COVID mortality

In the all-patient group, patients who experienced virologic rebound exhibited a significantly higher risk of post-acute mortality (HR, 1.52; 95% CI, 1.36–1.70; p value, <0.001) with a RD of 7.19% (95% CI, 4.88–9.51), compared with patients without virologic rebound (Figs. 2, 3). A similar increase in the risk of post-acute mortality was also observed in nirmatrelvir/ritonavir-treated patients (HR, 1.78; 95% CI, 1.41–2.25; p value, <0.001; RD, 12.55%) and molnupiravir-treated patients (HR, 1.47; 95% CI, 1.18–1.82; p value, <0.001; RD, 4.90%). The primary causes of death were listed in Supplementary Table 4.

Significant multiplicative interactions between nirmatrelvir/ritonavir received and virologic rebound on post-acute mortality was found (HR of multiplicative interaction: 0.68, 95% CI, 0.51–0.90; p value, 0.006; relative excess risk for interaction (RERI): −0.29, 95% CI, −0.88 to 0.29; p value, 0.33) (Supplementary Table 5). No significant interaction was found between using molnupiravir and virologic rebound on post-acute mortality (Supplementary Table 6).

## Post-COVID composite hospitalization and sequelae

In all patients group, a significant higher risk of composite hospitalization of post-COVID conditions was observed for patients with virologic rebound (HR, 1.22; 95% CI, 1.05–1.43; p value, 0.012) (Fig. 3). Specifically, virologic rebound is significantly associated with a greater risk of congestive heart failure (HR, 1.43; 95% CI, 1.08–1.90; p value, 0.013) and atrial fibrillation (HR, 1.41; 95% CI, 1.04–1.91; p value, 0.025). In the nirmatrelvir/ritonavir group, virologic rebound was found associated with a higher risk of composite hospitalization (HR, 1.50; 95% CI, 1.10–2.04; p value, 0.010), congestive heart failure (HR, 1.84; 95% CI, 1.02–3.34; p value, 0.044), atrial fibrillation (HR, 2.11; 95% CI, 1.25–3.56; p value, 0.005), and pancreatitis (HR, 8.49; 95% CI, 1.78–40.48; p value, 0.007). No significant results were found on post-acute sequelae in molnupiravir recipients.

The interaction analysis suggested that if the patients experienced virologic rebound, those receiving nirmatrelvir/ritonavir would have a lesser effect in reducing the risk of atrial fibrillation, compared to those without experiencing virologic rebound (HR of multiplicative

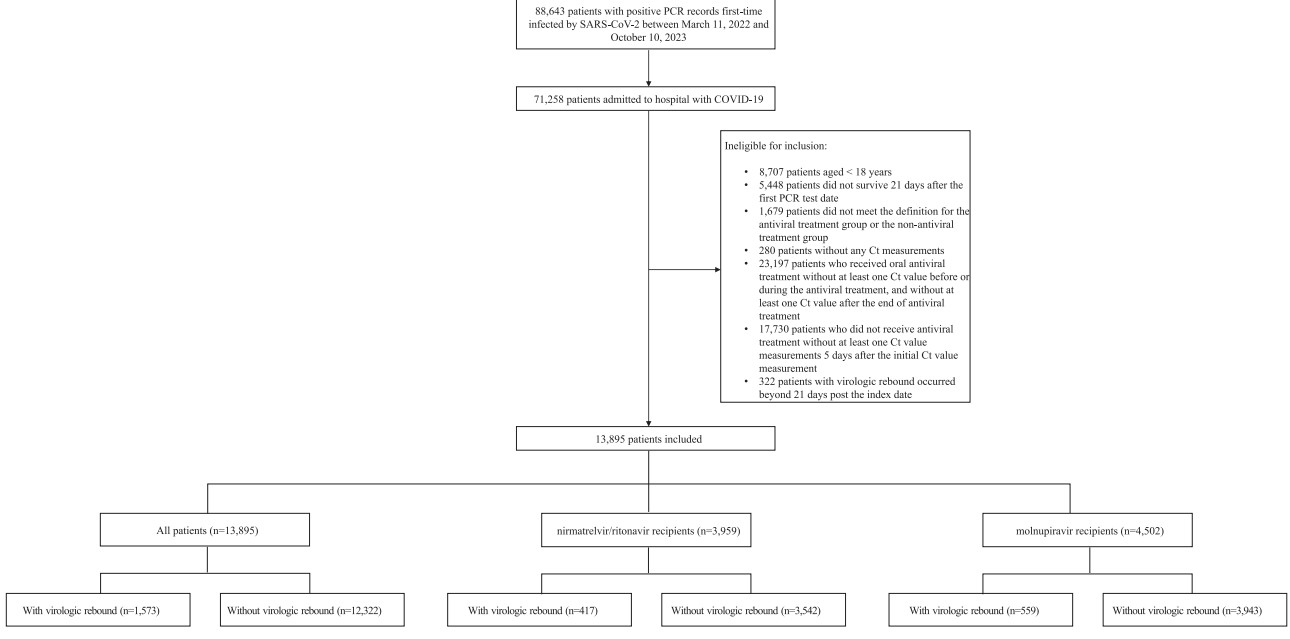

**Fig. 1 | Flowchart of patient inclusion and exclusion.**

**Table 1 | Baseline characteristics of patients with and without virologic rebound in all-patients, nirmatrelvir/ritonavir recipients, and molnupiravir recipients before weighting**

| | All patients | | | Nirmatrelvir/ritonavir recipients | | | Molnupiravir recipients | | |
|---|---|---|---|---|---|---|---|---|---|
| | Without VR (n = 12,322) | With VR (n = 1573) | SMD after weighting* | Without VR (n = 3542) | With VR (n = 417) | SMD after weighting* | Without VR (n = 3943) | With VR (n = 559) | SMD after weighting* |
| Age, years | 76.5 (15.6) | 74.6 (15.8) | -0.02 | 77.5 (14.2) | 77.9 (13.6) | -0.03 | 77.7 (15.1) | 73.9 (15.6) | -0.01 |
| Sex | | | 0.01 | | | 0.00 | | | 0.00 |
| Female | 5871 (47.6) | 659 (41.9) | | 1669 (47.1) | 184 (44.1) | | 2002 (50.8) | 238 (42.6) | |
| Male | 6451 (52.4) | 914 (58.1) | | 1873 (52.9) | 233 (55.9) | | 1941 (49.2) | 321 (57.4) | |
| Charlson Comorbidity Index | 0 (0-2) | 0 (0-2) | 0.01 | 0 (0-1) | 0 (0-1) | 0.02 | 0 (0-2) | 1 (0-2) | -0.01 |
| Myocardial infarction | 299 (2.4) | 68 (4.3) | | 37 (1.0) | 6 (1.4) | | 149 (3.8) | 42 (7.5) | |
| Congestive heart failure | 774 (6.3) | 149 (9.5) | | 122 (3.4) | 21 (5.0) | | 331 (8.4) | 74 (13.2) | |
| Peripheral vascular disease | 134 (1.1) | 28 (1.8) | | 24 (0.7) | 8 (1.9) | | 49 (1.2) | 11 (2.0) | |
| Cerebrovascular disease | 1086 (8.8) | 146 (9.3) | | 208 (5.9) | 27 (6.5) | | 473 (12.0) | 63 (11.3) | |
| Dementia | 316 (2.6) | 29 (1.8) | | 43 (1.2) | 2 (0.5) | | 163 (4.1) | 16 (2.9) | |
| Chronic pulmonary disease | 717 (5.8) | 92 (5.8) | | 153 (4.3) | 23 (5.5) | | 175 (4.4) | 19 (3.4) | |
| Rheumatic disease | 90 (0.7) | 16 (1.0) | | 23 (0.6) | 4 (1.0) | | 28 (0.7) | 8 (1.4) | |
| Peptic ulcer disease | 268 (2.2) | 37 (2.4) | | 64 (1.8) | 15 (3.6) | | 88 (2.2) | 10 (1.8) | |
| Mild liver disease | 396 (3.2) | 60 (3.8) | | 85 (2.4) | 12 (2.9) | | 146 (3.7) | 21 (3.8) | |
| Diabetes without complication | 1438 (11.7) | 208 (13.2) | | 301 (8.5) | 36 (8.6) | | 601 (15.2) | 99 (17.7) | |
| Diabetes with complication | 274 (2.2) | 47 (3.0) | | 42 (1.2) | 5 (1.2) | | 139 (3.5) | 27 (4.8) | |
| Hemiplegia or paraplegia | 99 (0.8) | 15 (1.0) | | 17 (0.5) | 1 (0.2) | | 41 (1.0) | 9 (1.6) | |
| Renal disease | 873 (7.1) | 192 (12.2) | | 85 (2.4) | 13 (3.1) | | 491 (12.5) | 127 (22.7) | |
| Malignancy | 947 (7.7) | 151 (9.6) | | 299 (8.4) | 54 (12.9) | | 245 (6.2) | 38 (6.8) | |
| Moderate-to-severe liver disease | 44 (0.4) | 7 (0.4) | | 11 (0.3) | 2 (0.5) | | 14 (0.4) | 3 (0.5) | |
| Metastatic solid tumor | 281 (2.3) | 46 (2.9) | | 92 (2.6) | 25 (6.0) | | 74 (1.9) | 10 (1.8) | |
| AIDS/HIV | 4 (<0.1) | 1 (0.1) | | 1 (<0.1) | 0 (<0.1) | | 2 (0.1) | 0 (0.0) | |
| Immunocompromised | 377 (3.1) | 80 (5.1) | -0.02 | 83 (2.3) | 17 (4.1) | 0.00 | 122 (3.1) | 25 (4.5) | -0.02 |
| Vaccination status | | | | | | | | | |
| 0 dose | 3269 (26.5) | 387 (24.6) | -0.02 | 772 (21.8) | 95 (22.8) | -0.01 | 947 (24.0) | 113 (20.2) | -0.04 |
| 1–2 doses | 3306 (26.8) | 434 (27.6) | 0.00 | 755 (21.3) | 100 (24.0) | 0.02 | 950 (24.1) | 143 (25.6) | -0.03 |
| ≥ 3 doses | 5747 (46.6) | 752 (47.8) | 0.01 | 2015 (56.9) | 222 (53.2) | 0.00 | 2046 (51.9) | 303 (54.2) | -0.05 |
| Concomitant treatment | | | | | | | | | |
| Dexamethasone | 4075 (33.1) | 621 (39.5) | -0.02 | 728 (20.6) | 140 (33.6) | -0.01 | 965 (24.5) | 133 (23.8) | -0.02 |
| Methylprednisolone | 21 (0.2) | 4 (0.3) | 0.00 | 4 (0.1) | 2 (0.5) | 0.01 | 1 (0.0) | 0 (0.0) | 0.00 |
| Prednisolone | 886 (7.2) | 157 (10.0) | -0.02 | 175 (4.9) | 31 (7.4) | -0.03 | 225 (5.7) | 45 (8.1) | -0.03 |
| Interferon | 66 (0.5) | 14 (0.9) | 0.00 | 4 (0.1) | 1 (0.2) | 0.00 | 9 (0.2) | 2 (0.4) | -0.01 |
| Baricitinib | 375 (3.0) | 100 (6.4) | -0.02 | 95 (2.7) | 36 (8.6) | -0.02 | 72 (1.8) | 11 (2.0) | -0.03 |
| Tocilizumab | 147 (1.2) | 31 (2.0) | 0.00 | 23 (0.6) | 8 (1.9) | 0.00 | 21 (0.5) | 10 (1.8) | 0.02 |
| Remdesivir | 2987 (24.2) | 526 (33.4) | 0.01 | 509 (14.4) | 101 (24.2) | -0.01 | 654 (16.6) | 113 (20.2) | 0.00 |
| Intensive care unit | 501 (4.1) | 119 (7.6) | 0.00 | 77 (2.2) | 31 (7.4) | 0.05 | 83 (2.1) | 19 (3.4) | -0.05 |
| Use of ventilation support | 299 (2.4) | 50 (3.2) | 0.00 | 54 (1.5) | 14 (3.4) | 0.01 | 47 (1.2) | 7 (1.3) | 0.00 |
| Initial Ct value | 22.3 (19.1-27.4) | 26.5 (21.4-32.2) | 0.08 | 21.4 (18.7-24.8) | 25.4 (20.2-30.5) | 0.08 | 21.0 (18.5-24.5) | 26.0 (21.0-31.6) | 0.08 |

**Table 1 (continued) | Baseline characteristics of patients with and without virologic rebound in all-patients, nirmatrelvir/ritonavir recipients, and molnupiravir recipients before weighting**

| | All patients | | | Nirmatrelvir/ritonavir recipients | | | Molnupiravir recipients | | |
|---|---|---|---|---|---|---|---|---|---|
| | Without VR (n = 12,322) | With VR (n = 1573) | SMD after weighting* | Without VR (n = 3542) | With VR (n = 417) | SMD after weighting* | Without VR (n = 3943) | With VR (n = 559) | SMD after weighting* |
| Duration between the first and the last Ct measurements during acute phase, days | 11 (8–16) | 8 (5–12) | −0.07 | 10 (7–15) | 9 (5–13) | 0.06 | 11 (8–16) | 8 (5–12) | 0.08 |
| Duration of the initial decline in Ct value for patients with VR[†] | - | 1 (1–2) | – | – | 2 (1–3) | – | – | 2 (1–3) | – |
| Duration between the index date and the date of observing VR[††] | - | 8 (4–13) | – | – | 8 (4–14) | – | – | 7 (4–12) | – |

Data are presented as mean (SD), median (IQR), or n (%).

*The standardized mean differences between patients with and without virologic rebound after weighting were presented.

[†]Initial decline: The first 2 samples showing initial decline in Ct values. To avoid transient dynamics of Ct values within a day, the Ct values were summarized as their mean value if a patient had multiple Ct measurements on the same day.

[††]The date of virologic rebound occurring was defined as the date of initial decline of 3 units in the Ct value.

VR virologic rebound, SMD standardized mean difference, Ct cycle threshold.

---

interaction: 0.39, 95% CI, 0.19–0.80; *p* value, 0.01; RERI, −1.49, 95% CI, −2.96 to −0.02; *p* value, 0.046) (Supplementary Tables 5, 6).

## Subgroup analyses

According to the subgroup analysis, the association between virologic rebound and post-acute death was comparable for those aged < 65 years and those aged ≥ 65 years, with HRs estimated at 1.73 (95% CI, 1.15–2.62; *p* value, 0.009) and 1.60 (95% CI, 1.42–1.81; *p* value, <0.001), respectively. With relatively fewer samples in the younger subgroup, no significant associations with post-acute sequelae were generally observed (Supplementary Fig. 1). The detrimental effect of virologic rebound incurred on post-acute death was more pronounced for patients with Charlson Comorbidity Index (CCI) ≥ 4 (HR, 2.03; 95% CI, 1.53–2.71; *p* value, <0.001) compared to those who had a CCI < 4 (HR, 1.56; 95% CI, 1.37–1.77; *p* value, <0.001) (Supplementary Fig. 2). Apart from that, virologic rebound was associated with a greater risk of post-acute mortality for vaccinated patients than (HR, 1.75; 95% CI, 1.51–2.02; *p* value, <0.001) for those unvaccinated (HR, 1.35; 95% CI, 1.10–1.66; *p* value, 0.004) (Supplementary Fig. 3).

## Sensitivity analyses

Our results are generally robust when the follow-up of post-COVID-19 outcomes was changed to 21–180 days or when the period of virologic rebound was shortened to 14 days post the index date, though the confidence interval became wider due to a smaller sample size (Supplementary Fig. 4 and Fig. 5). The main results had a minor change in terms of the post-acute sequelae when using the other three definitions of virologic rebound (Supplementary Fig. 6–8). When excluding patients who were admitted before their initial positive RT-PCR date, the effect sizes generally remained similar to that in the main analysis, although the confidence intervals became wider due to a reduced sample size (Supplementary Fig. 9). The post hoc analysis showed a significant association between virologic rebound and Ct measurements within 40 days after the index date in the three study groups (*p* value, <0.001) (Supplementary Fig. 10). The results of interaction analysis were robust when using another definition of virologic rebound (Supplementary Tables 7, 8).

## Discussion

In this study, we examined the association of SARS-CoV-2 rebound with post-acute outcomes in hospitalized patients with COVID-19 in Hong Kong during the Omicron epidemic, a period when most individuals experienced their first infection[12]. Our study primarily demonstrated a significant association between the virologic rebound and post-acute mortality. As inspired by Ghafari et al.[11], which showed that approximately 80% of persistent infections experienced rebounding viral dynamics, our study suggested that SARS-CoV-2 rebound may serve as an early virologic marker of persistent infections as well as post-acute conditions. The findings align with the study by Zuo et al.[10], which demonstrated a relationship between the persistence of residual SARS-CoV-2 RNA and long COVID symptoms, although Zuo et al. did not specifically evaluate virologic rebound. Nevertheless, Ghafari et al. reported that persistent infection was significantly related to long COVID at the first visit 12 weeks or longer, but not at 26 weeks or longer since the start of infection, likely due to a smaller sample size. Rather, our study employed real-world data with a larger sample size and longer follow-up time, which proved helpful in identifying significant associations with post-acute outcomes.

Our study demonstrated a link between SARS-CoV-2 rebound and post-COVID-19 mortality. Considering a connection between virologic rebound and persistent infections, the mechanism may be inferred from the potential impact of persistent infections on the pathophysiology of post-COVID-19 conditions[11,13,14]. One proposed direct mechanism is that virologic rebound indicates the presence of actively replicating virus during an infection, with a similar viral kinetics that

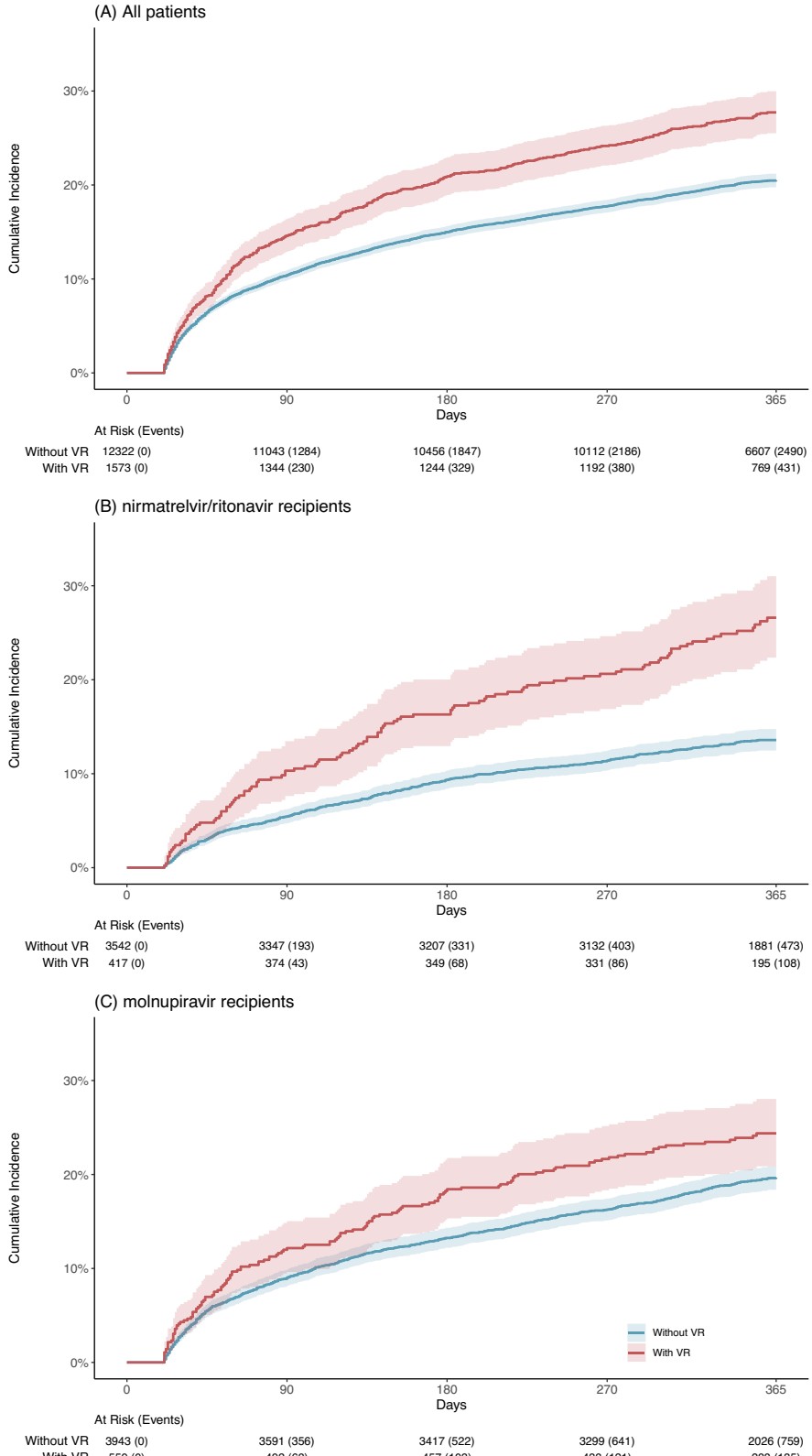

**Fig. 2 | Unadjusted cumulative incidence curve of post-acute mortality. A** All patients. **B** Nirmatrelvir/ritonavir recipients. **C** Molnupiravir recipients. Solid lines represent the cumulative incidence in patients with virologic rebound (red) and patients without virologic rebound (blue). Shaded regions indicate 95% confidence bands. Tables represent numbers at risk during the follow-up period in patients with and without virologic rebound. VR virologic rebound.

the initial rate of viral replication determines the course of the infection and infectious viral shedding[15]. Another possible mechanism is that individuals with virologic rebound may have lower levels and weaker production of receptor-binding domain IgA and IgG antibodies, especially among those with chronic conditions or immunocompromised status. This may lead to viral persistence, putting them at higher risk for post-COVID conditions[13,16] especially given studies showing a persistence of replication-competent virus within

**(A) All patients**

| Outcome | Without VR Events/Number at risk | With VR Events/Number at risk | Risk difference (95% CI) | HR (95% CI) | P value |
|---|---|---|---|---|---|
| Death | 2490/12322 | 431/1573 | 7.19% (4.88, 9.51) | 1.52 (1.36 to 1.70) | <0.001 |
| Composite hospitalization | 1439/9329 | 213/1102 | 3.9% (1.46, 6.35) | 1.22 (1.05 to 1.43) | 0.012 |
| Congestive heart failure | 406/11548 | 62/1425 | 0.84% (−0.28, 1.95) | 1.43 (1.08 to 1.90) | 0.013 |
| Atrial fibrillation | 362/11525 | 58/1448 | 0.86% (−0.19, 1.92) | 1.41 (1.04 to 1.91) | 0.025 |
| Coronary artery disease | 415/11490 | 63/1416 | 0.84% (−0.29, 1.96) | 1.23 (0.92 to 1.64) | 0.159 |
| Deep vein thrombosis | 76/12240 | 4/1560 | −0.36% (−0.65, −0.08) | 0.58 (0.21 to 1.61) | 0.297 |
| Chronic pulmonary disease | 240/11607 | 40/1483 | 0.63% (−0.23, 1.49) | 1.24 (0.86 to 1.79) | 0.253 |
| Acute respiratory distress syndrome | 428/12003 | 71/1506 | 1.15% (0.03, 2.27) | 1.12 (0.85 to 1.47) | 0.413 |
| Interstitial lung disease | 19/12297 | 5/1570 | 0.16% (−0.12, 0.45) | 1.53 (0.52 to 4.51) | 0.440 |
| Seizure | 113/12080 | 20/1528 | 0.37% (−0.22, 0.97) | 1.35 (0.81 to 2.24) | 0.253 |
| Anxiety | 13/12277 | 3/1564 | 0.09% (−0.14, 0.31) | 1.52 (0.40 to 5.83) | 0.542 |
| Post−traumatic stress disorder | 32/12241 | 5/1557 | 0.06% (−0.24, 0.35) | 0.74 (0.26 to 2.10) | 0.570 |
| End−stage renal disease | 58/12245 | 11/1563 | 0.23% (−0.2, 0.66) | 1.28 (0.62 to 2.60) | 0.504 |
| Acute kidney injury | 357/11990 | 53/1518 | 0.51% (−0.46, 1.49) | 1.24 (0.91 to 1.71) | 0.174 |
| Pancreatitis | 25/12279 | 8/1566 | 0.31% (−0.05, 0.67) | 2.39 (0.95 to 5.99) | 0.063 |

0.1 1 2 3 4
VR not at risk  VR at risk

**(B) nirmatrelvir/ritonavir recipients**

| Outcome | Without VR Events/Number at risk | With VR Events/Number at risk | Risk difference (95% CI) | HR (95% CI) | P value |
|---|---|---|---|---|---|
| Death | 473/3542 | 108/417 | 12.55% (8.19, 16.9) | 1.78 (1.41 to 2.25) | <0.001 |
| Composite hospitalization | 324/3009 | 57/329 | 6.56% (2.32, 10.79) | 1.50 (1.10 to 2.04) | 0.010 |
| Congestive heart failure | 76/3420 | 16/396 | 1.82% (−0.18, 3.82) | 1.84 (1.02 to 3.34) | 0.044 |
| Atrial fibrillation | 78/3463 | 22/397 | 3.29% (0.99, 5.59) | 2.11 (1.25 to 3.56) | 0.005 |
| Coronary artery disease | 83/3410 | 15/391 | 1.4% (−0.57, 3.38) | 1.32 (0.71 to 2.44) | 0.377 |
| Deep vein thrombosis | 14/3528 | 2/415 | 0.09% (−0.61, 0.78) | 1.03 (0.22 to 4.90) | 0.968 |
| Chronic pulmonary disease | 55/3390 | 8/394 | 0.41% (−1.05, 1.86) | 1.12 (0.51 to 2.44) | 0.773 |
| Acute respiratory distress syndrome | 73/3475 | 11/400 | 0.65% (−1.02, 2.32) | 1.14 (0.58 to 2.24) | 0.695 |
| Interstitial lung disease | 0/3540 | 0/417 | | | |
| Seizure | 19/3518 | 7/415 | 1.15% (−0.12, 2.41) | 2.88 (1.11 to 7.49) | 0.030 |
| Anxiety | 3/3530 | 1/414 | 0.16% (−0.33, 0.64) | 5.44 (0.53 to 55.36) | 0.153 |
| Post−traumatic stress disorder | 4/3526 | 0/416 | | | |
| End−stage renal disease | 6/3532 | 1/416 | 0.07% (−0.42, 0.56) | 1.16 (0.12 to 10.95) | 0.895 |
| Acute kidney injury | 77/3484 | 13/409 | 0.97% (−0.8, 2.74) | 1.46 (0.77 to 2.75) | 0.244 |
| Pancreatitis | 4/3528 | 3/417 | 0.61% (−0.21, 1.42) | 8.49 (1.78 to 40.48) | 0.007 |

0.1 1 2 3 4
VR not at risk  VR at risk

**(C) molnupiravir recipients**

| Outcome | Without VR Events/Number at risk | With VR Events/Number at risk | Risk difference (95% CI) | HR (95% CI) | P value |
|---|---|---|---|---|---|
| Death | 759/3943 | 135/559 | 4.9% (1.15, 8.66) | 1.47 (1.18 to 1.82) | <0.001 |
| Composite hospitalization | 430/2700 | 59/344 | 1.23% (−2.99, 5.44) | 1.18 (0.88 to 1.60) | 0.265 |
| Congestive heart failure | 153/3612 | 20/486 | −0.12% (−2, 1.76) | 1.33 (0.80 to 2.20) | 0.267 |
| Atrial fibrillation | 113/3503 | 16/500 | −0.03% (−1.68, 1.62) | 1.56 (0.89 to 2.76) | 0.123 |
| Coronary artery disease | 148/3560 | 20/471 | 0.09% (−1.85, 2.02) | 1.23 (0.74 to 2.06) | 0.425 |
| Deep vein thrombosis | 22/3904 | 2/550 | −0.2% (−0.76, 0.36) | 1.21 (0.28 to 5.33) | 0.801 |
| Chronic pulmonary disease | 66/3768 | 8/540 | −0.27% (−1.37, 0.83) | 0.73 (0.33 to 1.63) | 0.438 |
| Acute respiratory distress syndrome | 95/3847 | 19/537 | 1.07% (−0.57, 2.71) | 1.45 (0.83 to 2.55) | 0.193 |
| Interstitial lung disease | 2/3937 | 2/559 | 0.31% (−0.19, 0.81) | 6.16 (0.61 to 62.41) | 0.124 |
| Seizure | 45/3818 | 8/535 | 0.32% (−0.77, 1.4) | 1.15 (0.49 to 2.71) | 0.745 |
| Anxiety | 5/3919 | 2/554 | 0.23% (−0.28, 0.75) | 2.28 (0.37 to 13.97) | 0.374 |
| Post−traumatic stress disorder | 14/3906 | 2/550 | 0.01% (−0.53, 0.54) | 0.69 (0.13 to 3.68) | 0.666 |
| End−stage renal disease | 24/3902 | 2/551 | −0.25% (−0.81, 0.31) | 0.63 (0.14 to 2.84) | 0.551 |
| Acute kidney injury | 109/3795 | 19/525 | 0.75% (−0.94, 2.43) | 1.27 (0.73 to 2.23) | 0.401 |
| Pancreatitis | 8/3925 | 4/556 | 0.52% (−0.2, 1.23) | 1.76 (0.32 to 9.58) | 0.512 |

0.1 1 2 3 4
VR not at risk  VR at risk

**Fig. 3 | Effects of virologic rebound on each of the study outcomes. A** All patients. Cohort with virologic rebound (n = 1573) and cohort without virologic rebound (n = 12,322). **B** nirmatrelvir/ritonavir recipients. Cohort with virologic rebound (n = 417) and cohort without virologic rebound (n = 3542). **C** Molnupiravir recipients. Cohort with virologic rebound (n = 559) and cohort without virologic rebound (n = 3943). Adjusted HRs (square dots) and 95% CIs (error bars) are presented in (**A–C**). Adjustments were made for baseline covariates, including age, sex, Charlson Comorbidity Index, immunocompromised status, intensive care unit admission, concomitant treatments, ventilation support, COVID-19 vaccination status, the week of the index date, the initial Ct value at the index date, and the duration between the first and the last Ct measurement. The dashed vertical line in (**A–C**) represents the HR of 1.00. Statistical analysis with two-sided Wald test in (**A–C**). VR virologic rebound, CI confidence interval.

the body for extended periods, spanning months from the initial onset of infection[8,17]. The mechanism warrants further investigations to provide clarification.

Our study additionally demonstrated that the relationship between SARS-CoV-2 rebound and post-COVID-19 mortality is consistently observed across antiviral treatments provided to the patients, although the association with post-acute sequelae in molnupiravir recipients was not found. This finding builds upon previous studies that have suggested no association of virologic rebound with either nirmatrelvir/ritonavir or molnupiravir treatment, as well as acute COVID-19 outcomes[2,5–7], despite a potential variation due to treatment initiation time[18]. Of the post-acute sequelae, we observed a higher risk of post-acute atrial fibrillation and congestive heart failure, commonly reported post-COVID-19 conditions[19], in individuals with virologic rebound compared to those without. One speculation is that a prolonged viral persistence due to a deficient antiviral response may exacerbate cardiac function. Another potential explanation is that deprescribing medications for atrial fibrillation (e.g., amiodarone and dronedarone[20]) may be necessary for patients receiving nirmatrelvir/ritonavir to prevent adverse drug interactions. Noncompliance with the medications, combined with persistent virus presence in the body, may increase the risk of worsening cardiac function. While virologic rebound was shown to be related to seizure and pancreatitis among nirmatrelvir/ritonavir recipients, the low sample size remains a caution for interpretation. The observed association and underlying pathophysiological mechanisms warrant further verification.

Our subgroup analysis demonstrated that the risk of virologic rebound on post-acute death was more pronounced in individuals with more severe comorbidities. Given that these vulnerable populations may suffer from persistent infection with prolonged viral shedding[21,22], the virologic rebound, as an indicator of residual viral burden, likely heightened the risk of adverse post-acute outcomes. The variability in the relationship among these subgroups necessitates further large-sample studies for confirmation.

In this study, we focused on the hospitalized population with COVID-19, as they received close monitoring with regular viral burden assessments during their hospitalization. Consequently, their COVID-19 was generally more severe compared to the non-hospitalized patients, which has been reported in some other investigations[11,23]. Similarly, studies have shown variations in the rates of viral persistence between hospitalized patients[24] and household individuals[23]. However, it is important to note that our findings may not be fully generalized to patients with mild symptoms.

The major strength of this study is the use of real-world territory-wide inpatient data from all public hospitals, which covers more than 90% of all routine hospital admissions in Hong Kong. Close clinical monitoring during inpatient care facilitated thorough digital documentation of laboratory assessments, clinical records, and drug prescriptions, ensuring the validity of hospital admission information. Nevertheless, this study has notable limitations. Firstly, a lack of viral burden data after the acute infection period limited our investigation on the link between virologic rebound and viral persistence. Nevertheless, our post hoc analysis demonstrated an association between virologic rebound and higher post-acute viral burden up to 40 days post index date (Supplementary Fig. 10), likely supporting our postulation of the link between viral persistence and virologic rebound. Secondly, our study period was predominantly characterized by Omicron infections, specifically sub-lineages BA.5 and BA.2. Therefore, caution should be taken when extrapolating the study results to other variants and sub-lineages, as the duration of viral infection may vary. Thirdly, due to a lack of symptomatic data, we were unable to evaluate the symptom rebound. Fourthly, as abovementioned, patients with milder COVID-19 symptoms may not have undergone repeated viral burden monitoring, even if they were admitted to the hospital. This selection bias likely limited the generalizability of our findings to

severe patients. Apart from that, while we conducted a sensitivity analysis on different definitions of virologic rebound[6,9,25,26] and found our results to be robust, it should be noted that variations in the populations retrieved among the methods were observed, due to differences in testing frequency and assumptions regarding changes in Ct values. Fifthly, several post-acute sequelae outcomes do not have a sufficient number of events, leading to sparse data bias. Lastly, as this is an observational study, the design is subject to potential residual confounding such as healthcare-seeking behavior.

In conclusion, our study has demonstrated an association between the SARS-CoV-2 rebound and post-acute mortality in the hospitalized population, and this association is consistently found across antiviral treatment groups. The virologic rebound may serve as an early marker for post-COVID-19 condition, and the early identification enables healthcare providers to monitor and provide timely intervention for long COVID.

## Methods

### Study design and study setting

We conducted a population-wide retrospective cohort study using de-identified individual-level data of the hospitalized patients. The data were retrieved from the Hospital Authority (HA) and the Department of Health in Hong Kong. HA is a statutory body that provides public inpatient and outpatient services, serving over 7.3 million local citizens and accommodating more than 90% of all local hospitalizations. HA managed a centralized health record database, which contained routinely collected information on patients' demographic characteristics, death registry, hospitalization records, laboratory test records, and medication prescription records. With close clinical monitoring during inpatient care, the quality of the measurements of viral load data during inpatient care was ensured. These health records were further linked to the anonymized DH database, which provided population-based vaccination records. The diagnoses and procedures were coded according to the International Classification of Diseases, Ninth Revision, Clinical Modification (ICD-9-CM).

The study followed the STROBE (Strengthening the Reporting of Observational Studies in Epidemiology) reporting guideline. Ethics approval was obtained from the Joint CUHK-NTEC Clinical Research Ethics Committee (No. 2023.006). As this study was a retrospective analysis using secondary data without any personal information, the requirement for obtaining informed consent was waived and approved by the ethics committee.

### Study population

We included hospitalized patients aged 18 years or older who were first-time infected with SARS-CoV-2 and had positive reverse transcription polymerase chain reaction (RT-PCR) results from March 11, 2022 to October 10, 2023 (21 days before the end of data availability date, which is October 31, 2023). The SARS-CoV-2 Omicron variants were the dominant circulating variants during the study period. Patients who were admitted 3 days before or after the positive RT-PCR date were eligible for inclusion[27,28]. Such criteria also take into account the possible delay between case confirmation and hospital admission during a growth phase of the epidemic[27,28]. The index date was defined as the first recorded positive RT-PCR date, and the post-acute COVID-19 outcomes were assessed 21 days after the index date, a timeframe that was commonly adopted for evaluating the post-acute COVID-19 outcomes[28–30]. All included individuals had a Ct value at their index date.

Oral antiviral treatments, including nirmatrelvir/ritonavir (accessible to patients since March 6, 2022) and molnupiravir (accessible to patients since February 26, 2022), were dispensed to patients based on the HA COVID-19 patient management guidelines. According to the guidelines, patients who were at risk of progressing to severe COVID-19 were recommended to receive the antiviral treatments, such as

the elderly population, patients with asthma, chronic kidney disease, cancer, or diabetes mellitus. Patients who were prescribed nirmatrelvir/ritonavir and those who were prescribed molnupiravir within 5 days after the symptom onset (the index date was used as a proxy of the symptom onset date) were referred to as nirmatrelvir/ritonavir recipients and molnupiravir recipients, respectively. As treatment guidelines recommend using nirmatrelvir/ritonavir and molnupiravir within 5 days after illness onset, patients who used the antivirals after 5 days from the index date were excluded. Patients who used both nirmatrelvir/ritonavir and molnupiravir were also excluded.

The Ct value was obtained from the quantitative RT-PCR (RT-qPCR) assay. Serial Ct value measurements were obtained to determine virologic rebound. We included individuals who received oral antiviral treatment with at least one Ct value measurement before or during the antiviral treatment, and with at least one Ct value after the end of the treatment, with these conditions observed within the acute phase of infection—that is, 21 days after the index date. For patients who did not receive antiviral treatment, we excluded those without at least one Ct value five days after the initial Ct measurement[6], with these conditions observed within 21 days post the index date. We excluded patients with no Ct value measurements and those with virologic rebound (defined below) that occurred beyond 21 days after the index date. Ct values were retrieved for each patient after the index date. The Ct values were summarised as their mean value if a patient had multiple Ct measurements on the same day.

### Covariates
Baseline characteristics of recruited individuals were recorded, including age, sex, CCI computed based on diagnosis ascertained before the index date (Supplementary Table 1), immunocompromised status, the record of intensive care unit admission 21 days before the index date, initiation of concomitant treatments (dexamethasone, methylprednisolone, prednisolone, interferon-beta-1b, baricitinib, tocilizumab, and remdesivir) and ventilation support (including intubation, mechanical ventilation, and oxygen supplementation) within 21 days of the index date (Supplementary Table 2), COVID-19 vaccination status (unvaccinated, 1–2 doses, and ≥ 3 doses). The week of the index date and the initial Ct value measured at the index date were also used in weighting. Patients were considered as vaccinated if they had received the COVID-19 vaccine at least 14 days before the index date. The immunocompromised patients were those with diagnosed immunocompromising conditions (HIV, hematological malignancy, immune-mediated rheumatic disease, other hematological conditions, solid organ transplant, and bone marrow or stem cell transplant). Patients were also categorized as immunocompromised if they had a history of receiving or had remaining days supply of a monoclonal antibody within the last three months, an oral immunosuppressive drug within the last month, an oral glucocorticoid (equivalent to 20 mg/day of prednisone taken continuously) within the last month, or if they had received an immunosuppressive infusion or injection within the three months before the index date[30].

### Exposure and outcomes
The exposure group comprised patients who experienced SARS-CoV-2 virologic rebound within 21 days after the index date, defined as a decline in Ct value of at least 3 units between two consecutive Ct measurements, with such reduction persisting in at least one subsequent measurement, following the previous study[6]. Some other definitions of virologic rebound were based on the viral load measurements (viral mRNA copies), with at least a half to one-unit increase in $\log_{10}$-transformed viral load[4,8,9,25]. The Ct value from RT-qPCR assays was widely adopted as a proxy of viral load, with a lower Ct value indicating a higher viral load (an inverse correlation between log-transformed Ct value and viral load)[31–33]. A 3-point reduction in the Ct value is roughly equivalent to a tenfold increase in the viral load[34]. Ct

value was not available when a negative RT-qPCR result was observed and was imputed as 40, which was the limit of detection of the RT-qPCR assay. Other definitions of virologic rebound were tested in a sensitivity analysis.

The primary outcome was post-acute inpatient death, defined as a death that occurred 21–365 days post-index date. The secondary outcomes included post-acute composite hospitalization, defined as a hospitalization due to at least one of the 13 post-acute sequelae, including congestive heart failure, atrial fibrillation, coronary artery disease, deep vein thrombosis, chronic pulmonary disease, acute respiratory distress syndrome, interstitial lung disease, seizure, anxiety, post-traumatic stress disorder, end-stage renal disease, acute kidney injury, and pancreatitis[28,29] occurring 21–365 days post-index date (Supplementary Table 3). Hospitalization due to specific sequelae was also assessed[28,30,35,36]. To prevent any prior lingering conditions preceding the SARS-CoV-2 infection, individuals with a prior diagnosis of the condition of interest within three years before the index date were excluded from the analysis of post-acute sequelae. In this study, the sequelae of interest were primarily identified upon rehospitalization after discharge from the acute phase of SARS-CoV-2 infection, with a small proportion of patients with prolonged hospitalization also included. Individuals were followed from the index date until inpatient death, the occurrence of the clinical outcome events, 365 days after the index date, or the end date of data availability (October 31, 2023), whichever came first.

### Statistical analysis
SMR weighting was used to balance the baseline characteristics between patients with and without virologic rebound[37], with propensity scores estimated from multivariate logistic regression models. SMDs was applied as an indicator of covariates balance, with an SMD less than 0.1 indicating good balance between groups[38]. Unadjusted cumulative incidence of post-acute COVID-19 mortality was computed as a step function of follow-up time during the observational period, with RD between patients with and without virologic rebound computed. Cox proportional hazard models were used to estimate the HR between patients with and without virologic rebound in weighted cohorts. Robust standard errors were obtained through the Huber sandwich estimator. Data were analyzed for all patients and separately for nirmatrelvir/ritonavir recipients and molnupiravir recipients.

We conducted interaction analyses in the all-patient group to examine the effect modification of nirmatrelvir–ritonavir or molnupiravir on the association between virologic rebound and the post-acute outcomes. The RERI was calculated to evaluate the additive interaction[39], and the exponential of the coefficient of the product term of virologic rebound and antiviral treatments was obtained as the measurement of the multiplicative interaction.

Subgroup analyses were performed for the entire patient cohort according to age groups (< 65 years or ≥ 65 years), vaccination status (unvaccinated or ≥ 1 dose of vaccine), and the CCI categories (<4 or ≥4). Several sensitivity analyses were conducted: (1) the ascertainment window of post-acute COVID-19 outcomes was redefined from 21–365 days to 21–180 days after the index date; (2) patients with virologic rebound occurring beyond 14 days (instead of 21 days) after the index date were excluded; (3) restrict the analysis for patients admitted at the time or after the initial positive RT-PCR, which is considered as a proxy for admission related to COVID-19 symptomatology (4) Three alternative definitions of virologic rebound were evaluated based on previous studies: (i) a decrease in Ct value of at least 3 units after the end of oral antiviral treatment or treatment completion proxy within 21 days post the index date. The proxy of treatment completion date was defined for patients not receiving any oral antiviral treatment as 5 days after the index date, which was the median days between the index date and the treatment completion

date for antiviral recipients. This definition of virologic rebound accommodated the definitions applied in previous clinical trial and observational study[6,9]; (ii) at least two consecutive Ct measurements with values larger than or equal to 30 followed by at least two consecutive values less than 30[26]; (iii) a reduction in two consecutive Ct values from a value larger than 40 to a value less than or equal to 40[25]. In supporting the potential connection between virologic rebound and viral burden over time, a post-hoc analysis was conducted using the mixed effect models with outcome of Ct measurements within 40 days after the index date.

All analyses were conducted using R (version 4.2.2) (R Program for Statistical Computing).

## Reporting summary

Further information on research design is available in the Nature Portfolio Reporting Summary linked to this article.

## Data availability

The Hong Kong Hospital Authority and Department of Health, The Government of the Hong Kong Special Administrative Region, are the data custodians, and data requests to these parties can be made via email (hacpaaedr@ha.org.hk) and website (https://www.dh.gov.hk/english/aboutus/aboutus_pps/aboutus_pps.html), respectively. The cases' surveillance data and medication records were extracted from electronic records in the system managed by the Hong Kong Hospital Authority. The vaccine history was extracted from the COVID-19 surveillance database provided by the Department of Health in Hong Kong. Restrictions apply to the availability of these data, which were used under an agreement for the purposes of scientific research. The authors do not have the right to transfer or release the data, in whole or in part, and in whatever form or media, or to any other parties or place outside of Hong Kong, and must fully comply with the duties under the law relating to the protection of personal data, including those under the Personal Data (Privacy) Ordinance and its principles in all aspects.

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

## Acknowledgements

We thank the Hospital Authority and the Department of Health, Hong Kong Government for providing the data for this study. The Center for Health Systems and Policy Research funded by the Tung Foundation is acknowledged for the support throughout the conduct of this study. This research was funded by the Health and Medical Research Fund [EKY: grant numbers COVID190105, COVID19F03, INF-CUHK-1, COVID1903003], the RGC Collaborative Research Fund (CKPM: C6036-21GF), and the RGC theme-based research schemes (CKPM: T11-705/21-N). The funders of the study had no role in study design, data collection, data analysis, data interpretation, writing of the manuscript, or the decision to submit for publication.

## Author contributions

Study design and conceptualisation: Z.G., C.B., K.M.J., G.L., K.C.C. Data collection and pre-processing: Z.G., H.W., C.H.K.Y., T.Y.C., Y.Wei, E.K.Y. Data analysis and interpretation: Z.G., C.B., K.M.J., K.C.C. Writing—Original Draft: C.B., K.M.J., H.W., G.L., C.T.H., K.C.C. Writing—Review and Editing: Z.G., Y.W., H.W., C.H.K.Y., T.Y.C., X.J., C.L., S.Z., C.K.P.M., D.S.C.H., K.L., A.Y., E.K.Y. E.K.Y., and K.C.C. have accessed and verified all the data. All authors critically reviewed the manuscript and gave final approval for publication.

## Competing interests

The authors declare no competing interests.
