## [Transparent Peer Review file · Nature Communications]

SARS-CoV-2 rebound and post-acute mortality and hospitalization among patients admitted with COVID-19: cohort study

Corresponding Author: Dr Ka Chun Chong

Version 0:

Reviewer comments:

Reviewer #1

(Remarks to the Author)

Summary:

This is a real-world population-wide (>90% of local hospitals) adult retrospective cohort during the Omicron era in Hong Kong using robust statistical analyses. This analysis builds off a prior study by Wong et al published June 2023 to examine the association of viral load rebound among hospitalized patients and post-COVID-19 sequelae. In the current study, patients were categorized as having no antiviral treatment, receiving nirmatrelvir ritonavir, or receiving molnupiravir. The primary outcome was in-hospital death within 21-365 days after initial SARS-CoV-2 diagnosis, and secondary outcomes included 13 post-acute sequelae. Propensity score matching/weighting was done to balance the viral rebound (VR) and non-VR group followed by time to event analyses using a Cox proportional hazard model. There were many sensitivity analyses done using various definitions of viral load rebound and subgroup-analyses. The conclusion is that SARS-CoV-2 rebound in hospitalized patients may be a marker for subsequent mortality or other COVID-19 related complications. This is a well-done study with robust data evaluating outcomes in those with and without viral load rebound.

Key Results: Patients with virologic rebound had a significantly higher risk of death at 21-365 days compared to patients without virologic rebound.

Significance: Authors note that virologic rebound may serve as a marker for COVID-19 sequelae so healthcare providers could be more vigilant. However clinically I don't believe that quantitative PCR results are routinely available with SARS-CoV-2 so unclear how to identify patients with VL rebound or not (especially since symptom rebound is not often associated with VL rebound) to put this into practice.

Data Methodology: The statistical approach seems robust, but this is not my area of expertise.

References: The reference list is appropriate and comprehensive.

Major Points for Authors:

1) Generally, virologic rebound is discussed in reference to outpatients, and there have been several studies evaluating outpatient virologic rebound in relationship to oral antiviral use. The current study evaluates virologic rebound in hospitalized patients and then stratifies by treatment with an oral antiviral. It would be helpful to include in the methods why this group of hospitalized patients received oral antivirals. Was this based on provider discretion? What are the HA guidelines for oral antiviral treatment? Do all the patients receive oral antivirals or is it only if they have mild-moderate COVID-19? In that case, are they being hospitalized for another reason? This could be added to the methods. I would suggest including the term "in hospitalized patients" in the title as this is an important point to emphasize.

2) When I first read the title, I was expecting a manuscript on outpatient viral rebound and associations with Long COVID in the outpatient setting. In this study, however, the major outcome is COVID-19 mortality and other sequelae we would see in hospitalized patients. Perhaps the title could be reframed to include mortality and/or complications of COVID-19 in hospitalized patients. Also why does the title say "early" SARS-CoV-2 rebound? Is this referring to SARS-CoV-2 viral load rebound within 21 days? In the current literature, viral rebound has not been described much after 21 days of diagnosis/symptom start so perhaps the emphasis on "early" rebound is not needed.

3) One of the major theoretical links in the discussion is how virological rebound could be associated with persistent viral shedding and hence associated with post-COVID conditions. I would suggest spelling this out in the discussion a bit further. Perhaps individuals who experience virologic rebound are less able to suppress SARS-CoV-2 replication and hence may have prolonged viral shedding which could put them at higher risk for complications. Are there any studies to cite with a pathophysiological/immunological explanation? Were you able to examine persistent viral shedding in this analysis? Can you comment on prolonged viral shedding in participants with and without viral rebound? I think supplementary Table 9 gets at this (Line 341). Perhaps mentioning it in the methods would help clarify this hypothesized link between shedding and rebound.

4) In Line 55, the authors describe long COVID as health conditions after a patient has recovered from the acute phase. In the manuscript, it was not clear to me whether the patients with secondary complications (for example ARDS) had to have resolved from the acute COVID-19 hospitalization, be discharged, and then readmitted for this secondary complication. Or could they just have a prolonged hospitalization and then be diagnosed with a secondary complication after 21 days? Recommend clarification in the methods.

5) Could you comment on the duration of rebound? Was this a transient rise for 1-2 days between time points?

Minor Points:

1) Line 38 (Abstract): "We examined the association of early virologic rebound with post-acute mortality and hospitalization...." Is this re-hospitalization? Or any prolonged initial hospitalization?

2) Line 62: In this sentence, the authors note that approximately 1/3 of participants untreated had viral rebound. This estimate is on the higher side compared to other literature and refers to lower-level transient viral load rebound. There is much more variation in the literature on this estimate depending on the study methodology and frequency of testing (see Edelstein et al). I suggest noting that this is the estimate in one study, and there is a range depending on study definitions.

3) Line 68: Consider adding Edelstein et al to the citation number 9.

4) Line 112: Can you expand on this section? Nir-r and Molnupiravir were dispensed to patients based on the HA COVID-19 guidelines. Who was eligible for these medications per the guidelines? Was it only for mild-moderate COVID-19? What were the age and/or comorbidity requirements? Was this only in the outpatient setting prior to hospitalization or is this for hospitalized patients? (see major point above)

5) Line 165: change "the" to "a sensitivity analysis".

6) Line 172: Add 21-365 days to the sentence about secondary outcomes too.

7) Line 204: Change "occurred" to occurring.

8) Table 1 Line 534: Consider adding a row (or indicating in the text) the median number of days during the acute period (also what is the definition of the acute period days 1-21?) for which there was a CT value available? (This is helpful because the ability to detect viral load rebound may depend on the frequency of swabs available).

Reviewer #2

(Remarks to the Author)

This study by Chong et al. provides insights on the impact of an early virological rebound regarding the 1-year post infection sequelae, using a large national database from the department of health in Hong-Kong.

This study was conducted rigorously, with several sensitivity analyses corroborating the main finding: an increase in 1-year mortality among hospitalized patients experiencing an early virological rebound within the first 21 days after the index date

The main issues of this article revolve around the choice and definitions of virological rebound, which can compromise result interpretation. Additionally, the lack of detailed description of the virological data and the use of the index date as the symptom onset, coupled with inclusion criteria that may include non-COVID-related hospital admissions, raise concerns.

Below are the main comments for improving the manuscript :

- Regarding the main exposure, the authors use a definition of "a decline in Ct value at least 3 units between two consecutive Ct measurements with such reduction persisting in at least one subsequent measurement". However, this definition is less clear than the one they use as reference (<https://pmc.ncbi.nlm.nih.gov/articles/PMC9949892/>).

Could the authors clarify as whether the "consecutive" term they use refers to two samples taken one day apart (since the authors mention "daily CT values" in the study participants paragraph), and if not, add in the table 1 the median (IQR) delay between the 2 samples that define the initial decline in CT values ?

Furthermore, could the authors detail in table 1 the median number of samples by patients, as well as the median time since index date at which viral rebound was observed (e. g. time of first increase above 3 unit of CT as compared to the previous sample) ? This could be very impactful, especially regarding treatment arms, as viral rebounds are observed mainly after treatment cessation (<https://pmc.ncbi.nlm.nih.gov/articles/PMC10264150/> ; <https://pmc.ncbi.nlm.nih.gov/articles/PMC10312846/>)

Furthermore, I believe the most robust definition of viral rebound is the first alternative, using the treatment cessation timepoint (5 days post-index). While the authors did not find an interaction between treatment and viral rebound on mortality, supplementary table 4 reveals a significant multiplicative effect of N-R on death. Although the p-value is only 0.046 with the primary definition of viral rebound, this interaction should be explored using the other definitions of viral rebound. The results could have implications for N-R treatment protocols (see the impact of delayed treatment initiation on viral rebound: <https://pmc.ncbi.nlm.nih.gov/articles/PMC10312846/>).

- Regarding the use of index date as a proxy for the date of symptom onset and the inclusion of patients sampled after admission :

Viral dynamics of SARS-COV-2 are often analyzed using time since symptom onset as the starting point. However, the authors use here a time point that may differ significantly from the real time of symptom onset. As time of symptom onset is highly related to time of peak viral load (see notably [https://www.thelancet.com/journals/lanmic/article/PIIS2666-5247\(23\)00139-8/fulltext#fig1](https://www.thelancet.com/journals/lanmic/article/PIIS2666-5247(23)00139-8/fulltext#fig1)), a viral rebound could be wrongfully attributed to the natural increase of viral load if the initial PCR is sampled too early in the infection. The authors included patients who could be admitted up to 3 days prior to the first positive PCR, potentially for other medical reasons without presenting any symptoms. In addition, it is not known whether the PCR performed were related to COVID related symptoms or part of systematic screening during hospitalization.

To address this issue, I suggest two potential solutions :

- Report the percentage of patients initially admitted for COVID-19 related symptoms
- Perform sensitivity analyses separating patients admitted prior vs at the time/after the initial positive RT PCR, using the latter as a proxy for admission related to COVID-19 symptomatology.

Regarding the main outcome, could the authors report in supplementary appendix the causes of death ? As no other comorbidities seems affected by viral rebound in the main analysis, it could point the way towards risk factors unexplored in this study that could potentially be affected by viral rebound/long covid

Minor comment :

- Ref 4 points to <https://pmc.ncbi.nlm.nih.gov/articles/PMC10644265/> that asserts a different rate of virological rebound between untreated and N-R treated patients. The text refers to it as "incidence of virologic rebound is similar to those not receiving antivirals".

Reviewer #3

(Remarks to the Author)

In this manuscript, the authors examined the association of early virologic rebound with post-acute mortality and hospitalization due to post-acute sequelae among hospitalized patients with COVID-19. They extended the study to identify the relationship between virologic rebound and post-mortality in nirmatrelvir/ritonavir-treated and molnupiravir-treated patients. While identifying the virological rebound as an early marker for post-COVID-19 mortality and post-COVID-19 conditions is critical, the study hypothesis has not been presented based on scientific reasoning. Even though the number of data used is of large sample size, the relationship identified may misinform about the cause due to the lack of a scientific (biological) basis. Unless the biological basis is well described, the result is not scientifically significant to meet the standard for publication in Nature Communications.

Some of my concerns are:

1. The hypothesis "There is a relationship between SARS-CoV-2 rebound and post-COVID-19 conditions" needs to be based on biological facts. Describe with reference the biological mechanisms that motivated the hypothesis used. This is critical as the entire study and its conclusions are based on this hypothesis. Without strong biological motivation, the relation observed could be coincidence only.
2. The hospitalization and the secondary outcomes are considered due to at least one of the 13 post-acute sequelae. Why are those related to viral rebound and/or SARS-CoV-2 infection? Are there any biological causes?
3. Describe why those statistical analysis tools were used.
4. Should not treatments reduce viral rebound? Why are the risks observed similar in the treatment group?
5. The symptomatic individuals suffer significantly more from hospitalizations and deaths than asymptomatic. Do these asymptomatic individuals not have viral rebound?

Version 1:

Reviewer comments:

Reviewer #1

(Remarks to the Author)

The authors have adequately addressed all my initial comments. No further comments.

Reviewer #2

(Remarks to the Author)

I would like to thank the authors for their important work and the multiple analyses they added to the paper, which covered the points raised during the first round of review.

The only weakness that remains in this paper is the lack of information regarding date of symptom onset, that cannot be provided. Nonetheless, the sensitivity analysis of supplementary table 9 addresses this issue, without tackling it totally. Furthermore, this work sheds a light on the risk of sequelae post virological rebound, especially heart-related ones. This effect seemingly being driven by the use of nirmatrelvir/ritonavir during the acute phase of Covid-19 infection, it might be useful in guiding the use of antiviral drugs in patients with pre-existing heart diseases.

I do not have further comments to modify the article at this point, as it is ready for publication.

Reviewer #1 (Remarks to the Author):

Summary:

This is a real-world population-wide (>90% of local hospitals) adult retrospective cohort during the Omicron era in Hong Kong using robust statistical analyses. This analysis builds off a prior study by Wong et al published June 2023 to examine the association of viral load rebound among hospitalized patients and post-COVID-19 sequelae. In the current study, patients were categorized as having no antiviral treatment, receiving nirmatrelvir ritonavir, or receiving molnupiravir. The primary outcome was in-hospital death within 21-365 days after initial SARS-CoV-2 diagnosis, and secondary outcomes included 13 post-acute sequelae. Propensity score matching/weighting was done to balance the viral rebound (VR) and non-VR group followed by time to event analyses using a Cox proportional hazard model. There were many sensitivity analyses done using various definitions of viral load rebound and subgroup-analyses. The conclusion is that SARS-CoV-2 rebound in hospitalized patients may be a marker for subsequent mortality or other COVID-19 related complications. This is a well-done study with robust data evaluating outcomes in those with and without viral load rebound.

Key Results: Patients with virologic rebound had a significantly higher risk of death at 21-365 days compared to patients without virologic rebound.

Significance: Authors note that virologic rebound may serve as a marker for COVID-19 sequelae so healthcare providers could be more vigilant. However clinically I don't believe that quantitative PCR results are routinely available with SARS-CoV-2 so unclear how to identify patients with VL rebound or not (especially since symptom rebound is not often associated with VL rebound) to put this into practice.

Data Methodology: The statistical approach seems robust, but this is not my area of expertise.

References: The reference list is appropriate and comprehensive.

Response: Thanks for the positive response and constructive comments. We have substantially revised our manuscript following your suggestions.

Major Points for Authors:

1) Generally, virologic rebound is discussed in reference to outpatients, and there have been several studies evaluating outpatient virologic rebound in relationship to oral antiviral use. The current study evaluates virologic rebound in hospitalized patients and then stratifies by treatment with an oral antiviral. It would be helpful to include in the methods why this group of hospitalized patients received oral antivirals. Was this based on provider discretion? What are the HA guidelines for oral antiviral treatment? Do all the patients receive oral antivirals or is it only if they have mild-moderate COVID-19? In that case, are they being hospitalized for another reason? This could be added to the methods. I would suggest including the term "in hospitalized patients" in the title as this is an important point to emphasize.

Response: Thank you for the comment. The major reason of including hospitalized population in this study is that the measurements of viral loads require RT-qPCR assays that could only be used in the hospital settings. In the hospital settings in Hong Kong, close clinical monitoring during inpatient care could be ensured and it helps to facilitate comprehensive recording of laboratory assessments, clinical records, and drug prescriptions digitally. In addition, the hospitalization records in the public hospitals covers more than 90%

of all routine hospital admissions in Hong Kong, enhancing the representativeness of study population.

According to the HA guidelines for the clinical management of patients with COVID-19, patients who were at risk of progressing to severe COVID-19 were recommended to receive oral antiviral treatments such as elder population, patients with asthma, chronic kidney disease, cancer, or diabetes mellitus, no matter it is in outpatient or inpatient settings. Compared to outpatient individuals, we acknowledge that our inpatient population was generally elder and had more chronic conditions. Nevertheless, taking account for the abovementioned advantages of routine PCR testing in hospitals, we believe using the hospitalized patients with COVID-19 as our study population is preferable.

We have revised the title and the method section to justify why inpatient population was used and the details of antiviral prescription. A discussion has also been added.

Lines 1 (clean version):

“SARS-CoV-2 rebound and post-acute mortality and hospitalization among patients admitted to hospital with COVID-19: cohort study”

Lines 83:

“The data were retrieved from the Hospital Authority (HA) and the Department of Health in Hong Kong. HA is a statutory body that provides public inpatient and outpatient services, serving over 7.3 million local citizens and accommodating more than 90% of all local hospitalizations. HA managed a centralized health record database, which contained routinely collected information on patient’s demographic characteristics, death registry, hospitalization records, laboratory test records, and medication prescription records. With a close clinical monitoring during inpatient care, the quality for the measurements of viral load data during inpatient care was ensured.”

Lines 113:

“Oral antiviral treatments, including nirmatrelvir/ritonavir (accessible to patients since March 6, 2022) and molnupiravir (accessible to patients since February 26, 2022) were dispensed to patients based on the HA COVID-19 patient management guidelines. According to the guidelines, patients who were at risk of progressing to severe COVID-19 were recommended to receive the antiviral treatments such as elder population, patients with asthma, chronic kidney disease, cancer, or diabetes mellitus.”

Lines 366:

“In this study, we focused on the hospitalized population with COVID-19, as they received close monitoring with regular viral burden assessments during their hospitalization. Consequently, their COVID-19 was generally more severe compared to the non-hospitalized patients, which have been included in some other investigations [11, 33]. Similarly, studies have shown variations in the rates of viral persistence between hospitalized patients [34] and household participants [33]. However, it is important to note that our findings may not be fully generalized to patients with mild symptoms.”

2) When I first read the title, I was expecting a manuscript on outpatient viral rebound and associations with Long COVID in the outpatient setting. In this study, however, the major outcome is COVID-19 mortality and other sequelae we would see in hospitalized patients. Perhaps the title could be reframed to include mortality and/or complications of COVID-19 in hospitalized patients. Also why does the title say “early” SARS-CoV-2 rebound? Is this referring to SARS-CoV-2 viral load rebound within 21 days? In the current literature, viral rebound has not been described much after 21 days of diagnosis/symptom start so perhaps the emphasis on “early” rebound is not needed.

Response: Thank you for the comments. We have revised our title to specify clearer for the study outcomes and population. The original reason of using the term “early” is that we have defined a period of 21 days that was not been used in current literature for defining the virological rebound. As with your suggestion that viral rebound has not been described much after 21 days in literature, we have removed the term “early” in our title.

Lines 1:

“SARS-CoV-2 rebound and post-acute mortality and hospitalization among patients admitted to hospital with COVID-19: cohort study”

Moreover, we refined our inclusion and exclusion criteria for patients without viral rebound to overcome possible selection bias and make the study cohorts more comparable regarding the frequency and temporal distribution of RT-PCR testing (Originally, we only set this restriction to patients with viral rebound). We have now restricted the patients without virological rebound who meet the inclusion criteria of Ct measurements - that is, patients who received oral antiviral treatment with at least one Ct value before or during the antiviral treatment, and with at least one Ct value after the end of antiviral and patients who did not receive antiviral treatment and have at least one Ct value measurements 5 days after the initial Ct value measurement, **with such condition observed within 21 days since the index date.**

Lines 128:

“We included individuals who received oral antiviral treatment with at least one Ct value measurement before or during the antiviral treatment, and with at least one Ct value after the end of the treatment, with these conditions observed within the acute phase of infection—that was, 21 days after the index date. For patients who did not receive antiviral treatment, we excluded those without at least one Ct value five days after the initial Ct measurement [7], with these conditions observed within 21 days post the index date.”

The results have been updated and our conclusions remain robust to the revision.

3) One of the major theoretical links in the discussion is how virological rebound could be associated with persistent viral shedding and hence associated with post-COVID conditions. I would suggest spelling this out in the discussion a bit further. Perhaps individuals who experience virologic rebound are less able to suppress SARS-CoV-2 replication and hence may have prolonged viral shedding which could put them at higher risk for complications. Are there any studies to cite with a pathophysiological/immunological explanation? Were you able to examine persistent viral shedding in this analysis? Can you comment on prolonged viral shedding in participants with and without viral rebound? I think supplementary Table 9 gets at this (Line 341). Perhaps mentioning it in the methods would help clarify this hypothesized link between shedding and rebound.

Response: Thank you for the suggestion on the discussion about the biological mechanism. We suggested this hypothesis based on the relationship between viral rebound and persistent infections or prolonged viral shedding, as a risk factor of long COVID indicated by Zuo et al. and Ghafari et al. Several studies have demonstrated the link between virologic rebound and persistent infections [11, 26, 27]. One proposed direct mechanism is that the virologic rebound supports the presence of actively replicating virus during an infection, as with similar viral kinetic that the initial rate of viral replication determines the course of infection and infectious viral shedding [37]. Another possible mechanism is that individuals with viral rebound may have lower levels and weaker production of receptor-binding domain IgA and IgG antibodies, especially among those with chronic conditions or immunocompromised status, and thus resulting viral persistence that put them at higher risk for post-COVID conditions [26, 38], given that studies have showed a persistence of replication-competent virus within the body for extended periods, spanning months from the initial onset of infection [4, 28].

We have spelt out the theoretical links in the discussion:

Lines 325:

“To the best of our knowledge, this is the first study to demonstrate a link between SARS-CoV-2 rebound and post-COVID-19 mortality. Considering a connection between virologic rebound and persistent infections, the mechanism may be inferred from the potential impact of persistent infections on the pathophysiology of post-COVID-19 conditions [11, 26, 27]. One proposed direct mechanism is that virologic rebound indicates the presence of actively replicating virus during an infection, with a similar viral kinetics that the initial rate of viral replication determines the course of the infection and infectious viral shedding [37]. Another possible mechanism is that individuals with viral rebound may have lower levels and weaker production of receptor-binding domain IgA and IgG antibodies, especially among those with chronic conditions or immunocompromised status. This may lead to viral persistence, putting them at higher risk for post-COVID conditions [26, 38] especially given studies showing a persistence of replication-competent virus within the body for extended periods, spanning months from the initial onset of infection [4, 28]. The mechanism warrants further investigations to provide clarification.”

Due to insufficient monitoring of viral burden data 21-day acute infection period, we are unable to examine the effect of persistent viral shedding in this analysis. However, our study has conducted a post-hoc analysis using all the viral burden data beyond the 21-day period, and we discovered a significant association between virologic rebound and higher post-acute viral burden up to 40 days post index date (Supplementary Figure 9). While the results cannot be said as confirmatory, we believe the relationship likely supporting our postulation of the link between viral persistence and virologic rebound.

We have mentioned the post-hoc analysis in the revised manuscript:

Lines 227:

“In supporting the potential connection between virologic rebound and viral burden over time, a post-hoc analysis was conducted using the mixed effect models with outcome of Ct measurements within 40 days after the index date.”

Lines 303:

“The post-hoc analysis showed a significant association between virologic rebound and Ct measurements within 40 days after the index date in the three study groups (p-value, <0.001) (Supplementary Figure 9).”

Lines 379:

“Firstly, a lack of viral burden data after the acute infection period limited our investigation on the link between virologic rebound and viral persistence. Nevertheless, our post-hoc analysis demonstrated an association between virologic rebound and higher post-acute viral burden up to 40 days post index date (Supplementary Figure 9), likely supporting our postulation of the link between viral persistence and virologic rebound.”

Supplementary Figure 9. Daily Ct value in (A) all patients, (B) nirmatrelvir/ritonavir recipients, and (C) molnupiravir recipients with and without virological rebound within 40 days after the index date.

To compare the difference in Ct values between patients with and without virological rebound, p-values of the effect of virological rebound on Ct values were obtained using a generalized-additive mixed-effect model with random effect of patient-level intercept and fixed effects of covariates. Trend of Ct values throughout the 40 days post the index date was estimated by natural cubic spline with the knots set at 7 days and 17 days to aid in visualizing the pattern. The lines are predicted mean daily Ct values from the generalized-additive mixed-effect model.

4) In Line 55, the authors describe long COVID as health conditions after a patient has recovered from the acute phase. In the manuscript, it was not clear to me whether the patients with secondary complications (for example ARDS) had to have resolved from the acute COVID-19 hospitalization, be discharged, and then readmitted for this secondary complication. Or could they just have a prolonged hospitalization and then be diagnosed with a secondary complication after 21 days? Recommend clarification in the methods.

Response: Thank you for your comments. Our study defined the post-acute conditions as the conditions that were diagnosed 21 days after the index date, and we excluded participants with a prior history of these conditions within three years before the index date to avoid that a post-acute condition is just a carry-over condition before a SARS-CoV-2 infection. The sequelae of interest were those identified mainly upon rehospitalization after discharging from the acute phase of SARS-CoV-2 infection (i.e., 21 days), but we acknowledge a small proportion of cases (<20%) are the prolonged hospitalization with the conditions occurring at ≥ 21 -day post-index date. Our previous studies have also used these criteria to define the post-COVID outcomes e.g., [13,15,36].

We have revised the description of outcomes in the method section:

Lines 182:

“To prevent any prior lingering conditions preceding the SARS-CoV-2 infection, participants with a prior diagnosis of the condition of interest within three years before the index date were excluded from the analysis of post-acute sequelae. In this study, the sequelae of interest were mainly identified upon rehospitalization after discharging from the acute phase of SARS-CoV-2 infection, with a small proportion of patients with prolonged hospitalization also included.”

5) Could you comment on the duration of rebound? Was this a transient rise for 1-2 days between time points?

Response: Thank you for your comment. We defined the duration of rebound as the time interval during which the Ct values were at least three units lower than the initial Ct value and the median (IQR) of the duration of rebound was 4 (2-9) days. To avoid transient dynamics of Ct values within a day, we have summarised the Ct values as their mean value if a patient had multiple Ct measurements on the same day. The long-rebound period suggested that patients with viral rebound may have a prolonged viral shedding period compared with patients without viral rebound, which may explain the effect of viral rebound on the prognosis of COVID-19, given that persistent of within-host viral load was associated with long-COVID symptoms as reported by previous epidemiological and cross-sectional studies (Zuo et al; Ghafari et al).

Zuo W, He D, Liang C, Du S, Hua Z, Nie Q, et al. The persistence of SARS-CoV-2 in tissues and its association with long COVID symptoms: a cross-sectional cohort study in China. *Lancet Infect Dis.* 2024 Apr 22;S1473-3099(24)00171-3.

Ghafari M, Hall M, Golubchik T, Ayoubkhani D, House T, MacIntyre-Cockett G, et al. Prevalence of persistent SARS-CoV-2 in a large community surveillance study. *Nature.* 2024 Feb;626(8001):1094-1101.

Minor Points:

1) Line 38 (Abstract): “We examined the association of early virologic rebound with post-acute mortality and hospitalization...” Is this re-hospitalization? Or any prolonged initial hospitalization?

Response: Thanks for the comment. As the hospitalization outcomes included both re-hospitalization and prolonged hospitalization (though mainly rehospitalization), we keep using hospitalization in the abstract. In spite of it, we have revised the description of outcomes in the method section to clarify the definition (*see our response to comment 4*).

2) Line 62: In this sentence, the authors note that approximately 1/3 of participants untreated had viral rebound. This estimate is on the higher side compared to other literature and refers to lower-level transient viral load rebound. There is much more variation in the literature on this estimate depending on the study methodology and frequency of testing (see Edelstein et al). I suggest noting that this is the estimate in one study, and there is a range depending on study definitions.

Response: Thank you for the comment. We have noted it in the discussion:

Lines 392:

“...Apart from that, while we conducted a sensitivity analysis on different definitions of virologic rebound [7,9,16,24] and found our results to be robust, it should be noted that variations in the populations retrieved among the methods were observed, due to differences in testing frequency and assumptions regarding changes in Ct values.”

3) Line 68: Consider adding Edelstein et al to the citation number 9.

Response: Thanks for the comment and the reference has been included.

Lines 67:

“Nevertheless, inconsistent findings were found in several studies reporting more frequent virologic rebound in treated individuals e.g., [4,9].”

4) Line 112: Can you expand on this section? Nir-r and Molnupiravir were dispensed to patients based on the HA COVID-19 guidelines. Who was eligible for these medications per the guidelines? Was it only for mild-moderate COVID-19? What were the age and/or comorbidity requirements? Was this only in the outpatient setting prior to hospitalization or is this for hospitalized patients? (see major point above)

Response: Thanks for the comment and we have expanded this section. According to the HA guidelines for the clinical management of patients with COVID-19, patients who were at risk of progressing to severe COVID-19 were recommended to receive oral antiviral treatments such as patients with asthma, chronic kidney disease, cancer, or diabetes mellitus, no matter it is in outpatient or inpatient settings. Please see our above response and the revision in methods.

Lines 113:

“Oral antiviral treatments, including nirmatrelvir/ritonavir (accessible to patients since March 6, 2022) and molnupiravir (accessible to patients since February 26, 2022) were dispensed to patients based on the HA COVID-19 patient management

guidelines. According to the guidelines, patients who were at risk of progressing to severe COVID-19 were recommended to receive the antiviral treatments such as elder population, patients with asthma, chronic kidney disease, cancer, or diabetes mellitus.”

5) Line 165: change “the” to “a sensitivity analysis”.

Response: Thank you and we have revised it accordingly.

Lines 172:

“Other definitions of virologic rebound were tested in a sensitivity analysis.”

6) Line 172: Add 21-365 days to the sentence about secondary outcomes too.

Response: Thank you and we have revised it accordingly.

Lines 175:

“The secondary outcomes included post-acute composite hospitalization, defined as a hospitalization due to at least one of the 13 post-acute sequelae, including congestive heart failure, atrial fibrillation, coronary artery disease, deep vein thrombosis, chronic pulmonary disease, acute respiratory distress syndrome, interstitial lung disease, seizure, anxiety, post-traumatic stress disorder, end-stage renal disease, acute kidney injury, and pancreatitis [13,14] occurring 21-365 days post-index date (Supplementary Table 3). Hospitalization”

7) Line 204: Change “occurred” to occurring.

Response: Thank you and we have revised it accordingly.

Lines 214:

“(2) patients with virologic rebound occurring beyond 14 days (instead of 21 days) after the index date were excluded;”

8) Table 1 Line 534: Consider adding a row (or indicating in the text) the median number of days during the acute period (also what is the definition of the acute period days 1-21?) for which there was a CT value available? (This is helpful because the ability to detect viral load rebound may depend on the frequency of swabs available).

Response: Thank you for the comments. We defined the acute period as 21 days after the index date, in order to differentiating from post-COVID period. To provide more comprehensive information about the distributions of viral-rebound durations, we have followed the suggestion of reviewer 2 to add several rows of the duration statistics in Table 1, including duration between the first and the last Ct measurement during acute phase, duration between the initial decline (i.e., the first 2 samples) of Ct value for patients with VR, and duration between the index date and the date of observing VR.

Table 1. Baseline characteristics of patients with and without virological rebound in all-patients, nirmatrelvir/ritonavir recipients, and molnupiravir recipients before weighting.

	All patients		SMD after weighting*	Nirmatrelvir/ritonavir recipients		SMD after weighting*	Molnupiravir recipients		SMD after weighting*
	Without VR (n=12322)	With VR (n=1573)		Without VR (n=3542)	With VR (n=417)		Without VR (n=3943)	With VR (n=559)	
Age, years	76.5 (15.6)	74.6 (15.8)	-0.02	77.5 (14.2)	77.9 (13.6)	-0.03	77.7 (15.1)	73.9 (15.6)	-0.01
Sex			0.01			0.00			0.00
Female	5871 (47.6)	659 (41.9)		1669 (47.1)	184 (44.1)		2002 (50.8)	238 (42.6)	
Male	6451 (52.4)	914 (58.1)		1873 (52.9)	233 (55.9)		1941 (49.2)	321 (57.4)	
Charlson comorbidity index	0 (0-1)	0 (0-2)	0.01	0 (0-1)	0 (0-1)	0.02	0 (0-2)	1 (0-2)	-0.01
Myocardial infarction	299 (2.4)	68 (4.3)		37 (1.0)	6 (1.4)		149 (3.8)	42 (7.5)	
Congestive heart failure	774 (6.3)	149 (9.5)		122 (3.4)	21 (5.0)		331 (8.4)	74 (13.2)	
Peripheral vascular disease	134 (1.1)	28 (1.8)		24 (0.7)	8 (1.9)		49 (1.2)	11 (2.0)	
Cerebrovascular disease	1086 (8.8)	146 (9.3)		208 (5.9)	27 (6.5)		473 (12.0)	63 (11.3)	
Dementia	316 (2.6)	29 (1.8)		43 (1.2)	2 (0.5)		163 (4.1)	16 (2.9)	
Chronic pulmonary disease	717 (5.8)	92 (5.8)		153 (4.3)	23 (5.5)		175 (4.4)	19 (3.4)	
Rheumatic disease	90 (0.7)	16 (1.0)		23 (0.6)	4 (1.0)		28 (0.7)	8 (1.4)	
Peptic ulcer disease	268 (2.2)	37 (2.4)		64 (1.8)	15 (3.6)		88 (2.2)	10 (1.8)	
Mild liver disease	396 (3.2)	60 (3.8)		85 (2.4)	12 (2.9)		146 (3.7)	21 (3.8)	
Diabetes without complication	1438 (11.7)	208 (13.2)		301 (8.5)	36 (8.6)		601 (15.2)	99 (17.7)	
Diabetes with complication	274 (2.2)	47 (3.0)		42 (1.2)	5 (1.2)		139 (3.5)	27 (4.8)	
Hemiplegia or paraplegia	99 (0.8)	15 (1.0)		17 (0.5)	1 (0.2)		41 (1.0)	9 (1.6)	
Renal disease	873 (7.1)	192 (12.2)		85 (2.4)	13 (3.1)		491 (12.5)	127 (22.7)	
Malignancy	947 (7.7)	151 (9.6)		299 (8.4)	54 (12.9)		245 (6.2)	38 (6.8)	
Moderate-to-severe liver disease	44 (0.4)	7 (0.4)		11 (0.3)	2 (0.5)		14 (0.4)	3 (0.5)	
Metastatic solid tumor	281 (2.3)	46 (2.9)		92 (2.6)	25 (6.0)		74 (1.9)	10 (1.8)	
AIDS/HIV	4 (0.0)	1 (0.1)		1 (0.0)	0 (0.0)		2 (0.1)	0 (0.0)	
Immunocompromised	377 (3.1)	80 (5.1)	-0.02	83 (2.3)	17 (4.1)	0.00	122 (3.1)	25 (4.5)	-0.02
Vaccination status									
0 dose	3269 (26.5)	387 (24.6)	-0.02	772 (21.8)	95 (22.8)	-0.01	947 (24.0)	113 (20.2)	-0.04
1-2 doses	3306 (26.8)	434 (27.6)	0.00	755 (21.3)	100 (24.0)	0.02	950 (24.1)	143 (25.6)	-0.03
≥ 3 doses	5747 (46.6)	752 (47.8)	0.01	2015 (56.9)	222 (53.2)	0.00	2046 (51.9)	303 (54.2)	-0.05
Concomitant treatment									

Dexamethasone	4075 (33.1)	621 (39.5)	-0.02	728 (20.6)	140 (33.6)	-0.01	965 (24.5)	133 (23.8)	-0.02
Methylprednisolone	21 (0.2)	4 (0.3)	0.00	4 (0.1)	2 (0.5)	0.01	1 (0.0)	0 (0.0)	0.00
Prednisolone	886 (7.2)	157 (10.0)	-0.02	175 (4.9)	31 (7.4)	-0.03	225 (5.7)	45 (8.1)	-0.03
Interferon	66 (0.5)	14 (0.9)	0.00	4 (0.1)	1 (0.2)	0.00	9 (0.2)	2 (0.4)	-0.01
Baricitinib	375 (3.0)	100 (6.4)	-0.02	95 (2.7)	36 (8.6)	-0.02	72 (1.8)	11 (2.0)	-0.03
Tocilizumab	147 (1.2)	31 (2.0)	0.00	23 (0.6)	8 (1.9)	0.00	21 (0.5)	10 (1.8)	0.02
Remdesivir	2987 (24.2)	526 (33.4)	0.01	509 (14.4)	101 (24.2)	-0.01	654 (16.6)	113 (20.2)	0.00
Intensive care unit	501 (4.1)	119 (7.6)	0.00	77 (2.2)	31 (7.4)	0.05	83 (2.1)	19 (3.4)	-0.05
Use of ventilation support	299 (2.4)	50 (3.2)	0.00	54 (1.5)	14 (3.4)	0.01	47 (1.2)	7 (1.3)	0.00
Initial Ct value	22.3 (19.1-27.4)	26.5 (21.4-32.2)	0.08	21.4 (18.7-24.8)	25.4 (20.2-30.5)	0.08	21.0 (18.5-24.5)	26.0 (21.0-31.6)	0.08
Duration between the first and the last Ct measurement during acute phase, days	11 (8-16)	8 (5-12)	-0.07	10 (7-15)	9 (5-13)	0.06	11 (8-16)	8 (5-12)	0.08
Duration of the initial decline in Ct value for patients with VR*	-	1 (1-2)	-	-	2 (1-3)	-	-	2 (1-3)	-
Duration between the index date and the date of observing VR**	-	8 (4-13)	-	-	8 (4-14)	-	-	7 (4-12)	-

Data are presented as mean (SD), median (IQR), or n (%)

The standardized mean differences between patients with and without virological rebound after weighting were presented.

VR: virological rebound

SMD: standardized mean difference.

*Initial decline: The first 2 samples showing initial decline in CT values. To avoid transient dynamics of Ct values within a day, the Ct values were summarized as their mean value if a patient had multiple Ct measurements on the same day.

**The date of virological rebound occurring was defined as the date of initial decline of 3 units in the Ct value.

Reviewer #2 (Remarks to the Author):

This study by Chong et al. provides insights on the impact of an early virological rebound regarding the 1-year post infection sequelae, using a large national database from the department of health in Hong-Kong.

This study was conducted rigorously, with several sensitivity analyses corroborating the main finding: an increase in 1-year mortality among hospitalized patients experiencing an early virological rebound within the first 21 days after the index date

The main issues of this article revolve around the choice and definitions of virological rebound, which can compromise result interpretation. Additionally, the lack of detailed description of the virological data and the use of the index date as the symptom onset, coupled with inclusion criteria that may include non-COVID-related hospital admissions, raise concerns.

Below are the main comments for improving the manuscript:

1. Regarding the main exposure, the authors use a definition of "a decline in Ct value at least 3 units between two consecutive Ct measurements with such reduction persisting in at least one subsequent measurement". However, this definition is less clear than the one they use as reference (<https://pmc.ncbi.nlm.nih.gov/articles/PMC9949892/>). Could the authors clarify as whether the "consecutive" term they use refers to two samples taken one day apart (since the authors mention "daily CT values" in the study participants paragraph), and if not, add in the table 1 the median (IQR) delay between the 2 samples that define the initial decline in CT values?

Response: Thank you for your comment. Yes, the "consecutive" term refers to two samples not taken within a single day. To avoid transient dynamics of Ct values within a day, we have summarised the Ct values as their mean value if a patient had multiple Ct measurements on the same day. We have revised the definition in the methods:

Lines 136 (clean version):

"The Ct values were summarised as their mean value if a patient had multiple Ct measurements on the same day."

The duration of the initial decline in Ct values for the study cohorts was consistently short, with a median value of 1 to 2 days:

	All patients	Nirmatrelvir/ritonavir recipients	molnupiravir recipients
Duration of the initial decline in Ct value for patients with virologic rebound, median (IQR)	1 (1-2)	2 (1-3)	2 (1-3)

We have added these statistics in Table 1.

2. Furthermore, could the authors detail in table 1 the median number of samples by patients, as well as the median time since index date at which viral rebound was observed (e. g. time of first increase above 3 unit of CT as compared to the previous sample)? This could be very

impactful, especially regarding treatment arms, as viral rebounds are observed mainly after treatment cessation (<https://pmc.ncbi.nlm.nih.gov/articles/PMC10264150/> ; <https://pmc.ncbi.nlm.nih.gov/articles/PMC10312846/>)

Response: Thank you for your suggestion but we have a concern for reporting the statistics, given that we have restricted the minimum number of samples in our study population. Following the previous study by Wong et al, we only include patients that met the following criteria to ensure the frequency and temporal distribution of Ct measurements are comparable between patients with and without viral rebound:

- Patients who received oral antiviral treatment with at least one Ct value before or during the antiviral treatment, and with at least one Ct value after the end of antiviral treatment
- Patients who did not receive antiviral treatment and had at least one Ct value measurements 5 days after the initial Ct value measurement

Considering the potential selection bias (i.e., viral rebound would be more likely to be observed in severe patients beyond 21 days due to more frequent RT-PCR testing as compared with patients with mild symptoms), we also restricted the observational period for identifying viral rebound as within 21 days since the index date.

Because the number of samples in our study population have been restricted, the median numbers are actually quite similar and may bias the readers in reporting after our ‘restriction’. Rather, we believe that the potential bias may be resulted from the duration of the patient stay in the hospital (i.e., the longer time to stay, the more opportunity to do Ct measurements). Because of it, we prefer reporting the statistics of duration between the first and the last Ct measurement in each of the groups. In addition to this piece of information, we have also added the statistics of duration between the index date and the date of observing virological rebound, in order to assess whether treatment groups had a different duration compared to the non-treatment group.

Table 1 has been revised:

Table 1. Baseline characteristics of patients with and without virological rebound in all-patients, nirmatrelvir/ritonavir recipients, and molnupiravir recipients before weighting.

	All patients		SMD after weighting*	Nirmatrelvir/ritonavir recipients		SMD after weighting*	Molnupiravir recipients		SMD after weighting*
	Without VR (n=12322)	With VR (n=1573)		Without VR (n=3542)	With VR (n=417)		Without VR (n=3943)	With VR (n=559)	
Age, years	76.5 (15.6)	74.6 (15.8)	-0.02	77.5 (14.2)	77.9 (13.6)	-0.03	77.7 (15.1)	73.9 (15.6)	-0.01
Sex			0.01			0.00			0.00
Female	5871 (47.6)	659 (41.9)		1669 (47.1)	184 (44.1)		2002 (50.8)	238 (42.6)	
Male	6451 (52.4)	914 (58.1)		1873 (52.9)	233 (55.9)		1941 (49.2)	321 (57.4)	
Charlson comorbidity index	0 (0-1)	0 (0-2)	0.01	0 (0-1)	0 (0-1)	0.02	0 (0-2)	1 (0-2)	-0.01
Myocardial infarction	299 (2.4)	68 (4.3)		37 (1.0)	6 (1.4)		149 (3.8)	42 (7.5)	
Congestive heart failure	774 (6.3)	149 (9.5)		122 (3.4)	21 (5.0)		331 (8.4)	74 (13.2)	
Peripheral vascular disease	134 (1.1)	28 (1.8)		24 (0.7)	8 (1.9)		49 (1.2)	11 (2.0)	
Cerebrovascular disease	1086 (8.8)	146 (9.3)		208 (5.9)	27 (6.5)		473 (12.0)	63 (11.3)	
Dementia	316 (2.6)	29 (1.8)		43 (1.2)	2 (0.5)		163 (4.1)	16 (2.9)	
Chronic pulmonary disease	717 (5.8)	92 (5.8)		153 (4.3)	23 (5.5)		175 (4.4)	19 (3.4)	
Rheumatic disease	90 (0.7)	16 (1.0)		23 (0.6)	4 (1.0)		28 (0.7)	8 (1.4)	
Peptic ulcer disease	268 (2.2)	37 (2.4)		64 (1.8)	15 (3.6)		88 (2.2)	10 (1.8)	
Mild liver disease	396 (3.2)	60 (3.8)		85 (2.4)	12 (2.9)		146 (3.7)	21 (3.8)	
Diabetes without complication	1438 (11.7)	208 (13.2)		301 (8.5)	36 (8.6)		601 (15.2)	99 (17.7)	
Diabetes with complication	274 (2.2)	47 (3.0)		42 (1.2)	5 (1.2)		139 (3.5)	27 (4.8)	
Hemiplegia or paraplegia	99 (0.8)	15 (1.0)		17 (0.5)	1 (0.2)		41 (1.0)	9 (1.6)	
Renal disease	873 (7.1)	192 (12.2)		85 (2.4)	13 (3.1)		491 (12.5)	127 (22.7)	
Malignancy	947 (7.7)	151 (9.6)		299 (8.4)	54 (12.9)		245 (6.2)	38 (6.8)	
Moderate-to-severe liver disease	44 (0.4)	7 (0.4)		11 (0.3)	2 (0.5)		14 (0.4)	3 (0.5)	
Metastatic solid tumor	281 (2.3)	46 (2.9)		92 (2.6)	25 (6.0)		74 (1.9)	10 (1.8)	
AIDS/HIV	4 (0.0)	1 (0.1)		1 (0.0)	0 (0.0)		2 (0.1)	0 (0.0)	
Immunocompromised	377 (3.1)	80 (5.1)	-0.02	83 (2.3)	17 (4.1)	0.00	122 (3.1)	25 (4.5)	-0.02
Vaccination status									
0 dose	3269 (26.5)	387 (24.6)	-0.02	772 (21.8)	95 (22.8)	-0.01	947 (24.0)	113 (20.2)	-0.04
1-2 doses	3306 (26.8)	434 (27.6)	0.00	755 (21.3)	100 (24.0)	0.02	950 (24.1)	143 (25.6)	-0.03
≥ 3 doses	5747 (46.6)	752 (47.8)	0.01	2015 (56.9)	222 (53.2)	0.00	2046 (51.9)	303 (54.2)	-0.05
Concomitant treatment									

Dexamethasone	4075 (33.1)	621 (39.5)	-0.02	728 (20.6)	140 (33.6)	-0.01	965 (24.5)	133 (23.8)	-0.02
Methylprednisolone	21 (0.2)	4 (0.3)	0.00	4 (0.1)	2 (0.5)	0.01	1 (0.0)	0 (0.0)	0.00
Prednisolone	886 (7.2)	157 (10.0)	-0.02	175 (4.9)	31 (7.4)	-0.03	225 (5.7)	45 (8.1)	-0.03
Interferon	66 (0.5)	14 (0.9)	0.00	4 (0.1)	1 (0.2)	0.00	9 (0.2)	2 (0.4)	-0.01
Baricitinib	375 (3.0)	100 (6.4)	-0.02	95 (2.7)	36 (8.6)	-0.02	72 (1.8)	11 (2.0)	-0.03
Tocilizumab	147 (1.2)	31 (2.0)	0.00	23 (0.6)	8 (1.9)	0.00	21 (0.5)	10 (1.8)	0.02
Remdesivir	2987 (24.2)	526 (33.4)	0.01	509 (14.4)	101 (24.2)	-0.01	654 (16.6)	113 (20.2)	0.00
Intensive care unit	501 (4.1)	119 (7.6)	0.00	77 (2.2)	31 (7.4)	0.05	83 (2.1)	19 (3.4)	-0.05
Use of ventilation support	299 (2.4)	50 (3.2)	0.00	54 (1.5)	14 (3.4)	0.01	47 (1.2)	7 (1.3)	0.00
Initial Ct value	22.3 (19.1-27.4)	26.5 (21.4-32.2)	0.08	21.4 (18.7-24.8)	25.4 (20.2-30.5)	0.08	21.0 (18.5-24.5)	26.0 (21.0-31.6)	0.08
Duration between the first and the last Ct measurement during acute phase, days	11 (8-16)	8 (5-12)	-0.07	10 (7-15)	9 (5-13)	0.06	11 (8-16)	8 (5-12)	0.08
Duration of the initial decline in Ct value for patients with VR*	-	1 (1-2)	-	-	2 (1-3)	-	-	2 (1-3)	-
Duration between the index date and the date of observing VR**	-	8 (4-13)	-	-	8 (4-14)	-	-	7 (4-12)	-

Data are presented as mean (SD), median (IQR), or n (%)

The standardized mean differences between patients with and without virological rebound after weighting were presented.

VR: virological rebound

SMD: standardized mean difference.

*Initial decline: The first 2 samples showing initial decline in CT values. To avoid transient dynamics of Ct values within a day, the Ct values were summarized as their mean value if a patient had multiple Ct measurements on the same day.

**The date of virological rebound occurring was defined as the date of initial decline of 3 units in the Ct value.

Moreover, following the comment from reviewer 1, we have refined our inclusion and exclusion criteria for patients without virological rebound (i.e., restricting the pre-defined temporal distribution of Ct measurements to be observed within 21 days of the index date) to overcome possible selection bias and to make the study cohorts more comparable in terms of the frequency and distribution of RT-PCR testing. The inclusion criteria and the results have been updated.

Lines 128:

“We included individuals who received oral antiviral treatment with at least one Ct value measurement before or during the antiviral treatment, and with at least one Ct value after the end of the treatment, with these conditions observed within the acute phase of infection—that was, 21 days after the index date. For patients who did not receive antiviral treatment, we excluded those without at least one Ct value five days after the initial Ct measurement [7], with these conditions observed within 21 days post the index date.”

Wong CKH, Lau KTK, Au ICH, Lau EHY, Poon LLM, Hung IFN, Cowling BJ, Leung GM. Viral burden rebound in hospitalised patients with COVID-19 receiving oral antivirals in Hong Kong: a population-wide retrospective cohort study. *Lancet Infect Dis.* 2023 Jun;23(6):683-695.

3. Furthermore, I believe the most robust definition of viral rebound is the first alternative, using the treatment cessation timepoint (5 days post-index). While the authors did not find an interaction between treatment and viral rebound on mortality, supplementary table 4 reveals a significant multiplicative effect of N-R on death. Although the p-value is only 0.046 with the primary definition of viral rebound, this interaction should be explored using the other definitions of viral rebound. The results could have implications for N-R treatment protocols (see the impact of delayed treatment initiation on viral rebound:

<https://pmc.ncbi.nlm.nih.gov/articles/PMC10312846/>).

Response: Thank you for the suggestion. The updated results of the interaction analyses were shown in Supplementary Table 5-8 of the revised supplementary file. Under the primary definition of virological rebound, we found significant multiplicative interaction between the receipt of nirmatrelvir/ritonavir and virological rebound for death, all-cause hospitalization, atrial fibrillation, and pancreatitis (Supplementary Table 5).

Supplementary Table 5. Additive and multiplicative interaction analysis between nirmatrelvir/ritonavir use and virological rebound in all study participants for post-acute COVID-19 outcomes occurred 21-365 days after the index date

Outcome	RERI estimate	95% CI	P value	Multiplicative estimate	95% CI	P value
Death	-0.29	(-0.88, 0.29)	0.330	0.68	(0.51, 0.90)	0.006
Composite hospitalization	-0.54	(-1.26, 0.19)	0.146	0.63	(0.43, 0.92)	0.016
Congestive heart failure	-0.41	(-1.88, 1.06)	0.587	0.66	(0.32, 1.36)	0.260
Atrial fibrillation	-1.49	(-2.96, -0.02)	0.046	0.39	(0.19, 0.80)	0.010
Coronary artery disease	-0.20	(-1.35, 0.95)	0.736	0.79	(0.38, 1.64)	0.528
Deep vein thrombosis	NA	NA	NA	NA	NA	NA
Chronic pulmonary disease	0.72	(-0.71, 2.15)	0.322	1.27	(0.51, 3.14)	0.605
Acute respiratory distress syndrome	-0.08	(-1.44, 1.27)	0.906	0.87	(0.41, 1.85)	0.717
Interstitial lung disease	NA	NA	NA	NA	NA	NA
Seizure	-2.17	(-5.40, 1.06)	0.189	0.30	(0.08, 1.14)	0.078
Anxiety	NA	NA	NA	NA	NA	NA
Post-traumatic stress disorder	NA	NA	NA	NA	NA	NA
End-stage renal disease	3.37	(-4.01, 10.75)	0.371	1.27	(0.11, 14.70)	0.846
Acute kidney injury	-0.46	(-1.72, 0.79)	0.468	0.68	(0.31, 1.49)	0.337
Pancreatitis	-8.65	(-23.75, 6.45)	0.262	0.08	(0.01, 1.04)	0.053

RERI: relative excess risk for interaction. In this interaction analysis, the virological rebound independent variable was coded as 1 for patients with virological rebound and 0 for patients without virological rebound. The nirmatrelvir/ritonavir status was coded as 1 for not using nirmatrelvir/ritonavir and 0 for using nirmatrelvir/ritonavir. This coding was to ensure that the two independent variables represented risk factors instead of preventive factors, because preventive factors are not appropriate for the calculation of additive interaction unless recoded to risk factors [1]. The product term of the two independent variables was included in the Cox models. Relative excess risk for interaction (RERI) was calculated to evaluate the additive interaction between virological rebound and nirmatrelvir/ritonavir [2]. The exponential of the coefficient of the product term was obtained as the measurement of the multiplicative interaction.

There was no significant interaction between molnupiravir and virological rebound for the outcomes of interest (Supplementary Table 6).

Supplementary Table 6. Additive and multiplicative interaction analysis between molnupiravir use and virological rebound in all study participants for post-acute COVID-19 outcomes occurred 21-365 days after the index date

Outcome	RERI estimate	95% CI	P value	Multiplicative estimate	95% CI	P value
Death	0.20	(-0.21, 0.61)	0.339	1.01	(0.78, 1.30)	0.951
Composite hospitalization	0.08	(-0.40, 0.56)	0.738	1.01	(0.70, 1.46)	0.954
Congestive heart failure	0.21	(-0.59, 1.01)	0.613	1.17	(0.61, 2.24)	0.637
Atrial fibrillation	-0.36	(-1.45, 0.73)	0.519	0.73	(0.35, 1.53)	0.410
Coronary artery disease	0.07	(-0.75, 0.89)	0.867	1.02	(0.53, 1.96)	0.942
Deep vein thrombosis	NA	NA	NA	NA	NA	NA
Chronic pulmonary disease	1.15	(-0.06, 2.36)	0.062	2.00	(0.81, 4.94)	0.135
Acute respiratory distress syndrome	0.15	(-1.02, 1.31)	0.807	0.89	(0.47, 1.68)	0.724
Interstitial lung disease	-2.62	(-15.09, 9.85)	0.680	0.25	(0.02, 3.52)	0.306
Seizure	-0.25	(-1.38, 0.89)	0.671	0.77	(0.22, 2.62)	0.672
Anxiety	NA	NA	NA	NA	NA	NA
Post-traumatic stress disorder	0.31	(-1.55, 2.18)	0.741	1.40	(0.17, 11.65)	0.753
End-stage renal disease	1.49	(-0.35, 3.34)	0.113	3.37	(0.60, 18.88)	0.167
Acute kidney injury	-0.16	(-1.07, 0.76)	0.734	0.86	(0.42, 1.76)	0.686
Pancreatitis	-2.68	(-8.01, 2.66)	0.325	0.20	(0.02, 2.48)	0.211

RERI: relative excess risk for interaction. In this interaction analysis, the virological rebound independent variable was coded as 1 for patients with virological rebound and 0 for patients without virological rebound. The molnupiravir status was coded as 1 for not using molnupiravir and 0 for using molnupiravir. This coding was to ensure that the two independent variables represented risk factors instead of preventive factors, because preventive factors are not appropriate for the calculation of additive interaction unless recoded to risk factors [1]. The product term of the two independent variables was included in the Cox models. Relative excess risk for interaction (RERI) was calculated to evaluate the additive interaction between virological rebound and molnupiravir [2]. The exponential of the coefficient of the product term was obtained as the measurement of the multiplicative interaction.

We have followed your suggestion to perform an interaction analysis using the first alternative, whereby a decrease in Ct value of at least 3 units after the end of oral antiviral treatment or treatment completion proxy within 21 days post the index date. The results showed a similar pattern in the effect sizes and statistical significance. (Supplementary Table 7-8).

Supplementary Table 7. Additive and multiplicative interaction analysis between nirmatrelvir/ritonavir use and virological rebound in all study participants for post-acute COVID-19 outcomes occurred 21-365 days after the index date, **using alternative definition of virological rebound (i): a decrease in Ct value of at least 3 units after the end of oral antiviral treatment or treatment completion proxy within 21 days post the index date of virological rebound.**

Outcome	RERI estimate	95% CI	P value	Multiplicative estimate	95% CI	P value
Death	-0.54	(-0.98, -0.09)	0.019	0.62	(0.50, 0.78)	<0.001
Composite hospitalization	-0.50	(-1.03, 0.03)	0.065	0.67	(0.50, 0.90)	0.007
Congestive heart failure	-0.06	(-0.89, 0.77)	0.891	0.92	(0.49, 1.72)	0.794
Atrial fibrillation	-0.98	(-2.09, 0.14)	0.086	0.51	(0.28, 0.90)	0.022
Coronary artery disease	-0.33	(-1.41, 0.74)	0.544	0.68	(0.40, 1.17)	0.163
Deep vein thrombosis	0.21	(-1.65, 2.06)	0.826	1.37	(0.23, 8.05)	0.729
Chronic pulmonary disease	-0.33	(-1.50, 0.85)	0.587	0.74	(0.37, 1.49)	0.399
Acute respiratory distress syndrome	-0.40	(-1.43, 0.63)	0.445	0.79	(0.44, 1.40)	0.411
Interstitial lung disease	NA	NA	NA	NA	NA	NA
Seizure	0.90	(-0.89, 2.70)	0.324	1.58	(0.43, 5.75)	0.487
Anxiety	NA	NA	NA	NA	NA	NA
Post-traumatic stress disorder	NA	NA	NA	NA	NA	NA
End-stage renal disease	-2.70	(-10.95, 5.56)	0.522	0.33	(0.05, 2.19)	0.253
Acute kidney injury	-0.44	(-1.41, 0.52)	0.365	0.68	(0.38, 1.22)	0.195
Pancreatitis	-0.74	(-3.98, 2.51)	0.657	0.56	(0.06, 5.42)	0.614

RERI: relative excess risk for interaction. In this interaction analysis, the virological rebound independent variable was coded as 1 for patients with virological rebound and 0 for patients without virological rebound. The nirmatrelvir/ritonavir status was coded as 1 for not using nirmatrelvir/ritonavir and 0 for using nirmatrelvir/ritonavir. This coding was to ensure that the two independent variables represented risk factors instead of preventive factors, because preventive factors are not appropriate for the calculation of additive interaction unless recoded to risk factors [1]. The product term of the two independent variables was included in the Cox models. Relative excess risk for interaction (RERI) was calculated to evaluate the additive interaction between virological rebound and nirmatrelvir/ritonavir [2]. The exponential of the coefficient of the product term was obtained as the measurement of the multiplicative interaction.

Supplementary Table 8. Additive and multiplicative interaction analysis between molnupiravir use and virological rebound in all study participants for post-acute COVID-19 outcomes occurred 21-365 days after the index date, **using alternative definition of virological rebound (i): a decrease in Ct value of at least 3 units after the end of oral antiviral treatment or treatment completion proxy within 21 days post the index date of virological rebound.**

Outcome	RERI estimate	95% CI	P value	Multiplicative estimate	95% CI	P value
Death	-0.09	(-0.39, 0.21)	0.551	0.87	(0.71, 1.06)	0.173
Composite hospitalization	-0.03	(-0.37, 0.32)	0.878	0.95	(0.72, 1.26)	0.743
Congestive heart failure	0.03	(-0.47, 0.53)	0.906	1.05	(0.63, 1.74)	0.858
Atrial fibrillation	-0.30	(-1.04, 0.43)	0.415	0.77	(0.45, 1.31)	0.332
Coronary artery disease	0.13	(-0.51, 0.76)	0.698	1.05	(0.65, 1.70)	0.830
Deep vein thrombosis	-1.10	(-3.37, 1.18)	0.345	0.47	(0.13, 1.66)	0.239
Chronic pulmonary disease	0.38	(-0.58, 1.35)	0.435	1.28	(0.64, 2.56)	0.483
Acute respiratory distress syndrome	-0.39	(-1.35, 0.56)	0.418	0.73	(0.45, 1.21)	0.227
Interstitial lung disease	-0.59	(-13.40, 12.21)	0.928	0.28	(0.02, 3.22)	0.304
Seizure	0.66	(-0.17, 1.49)	0.122	2.13	(0.79, 5.74)	0.136
Anxiety	1.00	(-1.12, 3.12)	0.355	3.37	(0.19, 58.41)	0.403
Post-traumatic stress disorder	1.19	(-0.79, 3.17)	0.239	2.88	(0.53, 15.59)	0.219
End-stage renal disease	0.62	(-0.38, 1.63)	0.225	3.44	(0.63, 18.71)	0.153
Acute kidney injury	-0.15	(-0.88, 0.58)	0.685	0.85	(0.50, 1.44)	0.547
Pancreatitis	-4.63	(-11.82, 2.56)	0.207	0.16	(0.02, 1.09)	0.062

RERI: relative excess risk for interaction. In this interaction analysis, the virological rebound independent variable was coded as 1 for patients with virological rebound and 0 for patients without virological rebound. The molnupiravir status was coded as 1 for not using molnupiravir and 0 for using molnupiravir. This coding was to ensure that the two independent variables represented risk factors instead of preventive factors, because preventive factors are not appropriate for the calculation of additive interaction unless recoded to risk factors [1]. The product term of the two independent variables was included in the Cox models. Relative excess risk for interaction (RERI) was calculated to evaluate the additive interaction between virological rebound and molnupiravir [2]. The exponential of the coefficient of the product term was obtained as the measurement of the multiplicative interaction.

In general, the interaction analyses suggest if the patients experienced virologic rebound, those receiving nirmatrelvir/ritonavir would have a lesser effect in reducing the risk of post-acute outcomes, compared to those without experiencing virologic rebound.

Please see the corresponding revision in the manuscript:

Lines 260:

“Significant multiplicative interactions between nirmatrelvir/ritonavir received and virologic rebound on post-acute mortality was found (HR of multiplicative interaction: 0.68, 95% CI, 0.51–0.90; p-value, 0.006; RERI: -0.29, 95% CI, -0.88–0.29; p-value, 0.33) (Supplementary Table 5). No significant interaction was found between using molnupiravir and virologic rebound on post-acute mortality (Supplementary Table 6).”

Lines 278:

“The interaction analysis suggested that if the patients experienced virologic rebound, those receiving nirmatrelvir/ritonavir would have a lesser effect in reducing the risk of atrial fibrillation, compared to those without experiencing virologic rebound (HR of multiplicative interaction: 0.39, 95% CI, 0.19–0.80; p-value, 0.01; RERI: -1.49, 95% CI, -2.96– -0.02; p-value, 0.046) (Supplementary Table 5 and Table 6).”

Lines 306:

“The results of interaction analysis were robust when using another definition of virologic rebound (Supplementary Table 7 and Table 8).”

Supporting by your suggested reference, we have also discussed the potential impact of treatment initiation time on the viral rebound.

Lines 343:

“This finding builds upon previous studies that have suggested no association of virologic rebound with either nirmatrelvir/ritonavir or molnupiravir treatment, as well as acute COVID-19 outcomes [2, 6-8], despite a potential variation due to treatment initiation time [39].”

4. Regarding the use of index date as a proxy for the date of symptom onset and the inclusion of patients sampled after admission:

Viral dynamics of SARS-COV-2 are often analyzed using time since symptom onset as the starting point. However, the authors use here a time point that may differ significantly from the real time of symptom onset. As time of symptom onset is highly related to time of peak viral load (see notably [https://www.thelancet.com/journals/lanmic/article/PIIS2666-5247\(23\)00139-8/fulltext#fig1](https://www.thelancet.com/journals/lanmic/article/PIIS2666-5247(23)00139-8/fulltext#fig1)), a viral rebound could be wrongfully attributed to the natural increase of viral load if the initial PCR is sampled too early in the infection. The authors included patients who could be admitted up to 3 days prior to the first positive PCR, potentially for other medical reasons without presenting any symptoms. In addition, it is not known whether the PCR performed were related to COVID related symptoms or part of systematic screening during hospitalization.

To address this issue, I suggest two potential solutions:

- Report the percentage of patients initially admitted for COVID-19 related symptoms
- Perform sensitivity analyses separating patients admitted prior vs at the time/after the initial positive RT PCR, using the latter as a proxy for admission related to COVID-19 symptomatology.

Response: Thank you for your suggestions. We are sorry that the information of symptom onset time is unavailable in our dataset, given that it was not compulsory for the patients to report their onset time in the system. According to the guideline in Hospital Authority, patients were advised to receive PCR testing during the inpatient care if they did not have any PCR testing before an admission.

Following your suggestion, we have conducted sensitivity analysis by only including patients admitted at the time/after the initial positive RT-PCR, which is used as a proxy for admission related to COVID-19 symptomatology. The results were shown in Supplementary Figure 9 of revised supplementary file.

Supplementary Figure 9. Sensitivity analysis of the association between virological rebound and each post-acute COVID-19 outcome in patients who were admitted at the time/after the initial positive RT-PCR. (A) all patients, (B) nirmatrelvir/ritonavir recipients, and (C) molnupiravir recipients for post-acute COVID-19 outcomes 21-365 days after the index date.

The effect sizes generally remained similar to that in the main analysis, although the confidence intervals became wider due to a reduced sample size.

In our study, the inclusion of patients whose admission date was up to 3 days before the initial positive RT-PCR date was intended to accommodate potential delays in confirmation of SARS-CoV-2 infection during surges of COVID-19 cases observed in the Omicron outbreaks. This inclusion criteria was also adopted by Wong et al., and Wang et al., (work done by our team) in studies investigating the real-world effectiveness of nirmatrelvir/ritonavir against acute infection and post-acute sequelae, respectively.

Lines 107:

“Such criteria also take into account the possible delay between case confirmation and hospital admission during a growth phase of the epidemic [12,13].”

5. Regarding the main outcome, could the authors report in supplementary appendix the causes of death? As no other comorbidities seems affected by viral rebound in the main analysis, it could point the way towards risk factors unexplored in this study that could potentially be affected by viral rebound/long covid

Response: Thank you for your suggestion. We provided an additional table listing the primary cause of death among included participants, as shown in Supplementary Table 4 in the supplementary file.

Supplementary Table 4. The principal cause of death of study participants during the observational period.

	Number of Patients (n=2897*)
Disease category [#] , n (%)	
diseases of the blood and blood-forming organs	21 (0.7)
diseases of the circulatory system	256 (8.8)
diseases of the digestive system	123 (4.2)
diseases of the genitourinary system	132 (4.6)
diseases of the musculoskeletal system and connective tissue	13 (0.4)
diseases of the nervous system and sense organs	7 (0.2)
diseases of the respiratory system	1536 (53.0)
diseases of the skin and subcutaneous tissue	39 (1.3)
endocrine, nutritional and metabolic diseases, and immunity disorders	31 (1.1)
factors influencing health status and contact with health services	8 (0.3)
infectious and parasitic diseases	108 (3.7)
injury and poisoning	42 (1.4)
mental disorders	31 (1.1)
neoplasms	371 (12.8)
symptoms, signs, and ill-defined conditions	179 (6.2)

[#] the disease category for the principle cause of death is referenced from World Health Organization, Manual of the International Classification of Diseases, Injuries, and Causes of Death, Ninth Revision. Geneva: World Health Organization, 1977. [cited 2025 Mar 13].

Available from:

https://web.archive.org/web/20191228233120/https://simba.isr.umich.edu/restricted/docs/Mortality/icd_09_codes.pdf

*The cause of death record of 24 patients is not available in the dataset.

Minor comment:

- Ref 4 points to <https://pmc.ncbi.nlm.nih.gov/articles/PMC10644265/> that asserts a different rate of virological rebound between untreated and N-R treated patients. The text refers to it as "incidence of virological rebound is similar to those not receiving antivirals".

Response: Thank you for identifying our misinterpretation. We incorrectly cited this reference in the first part of our paragraph. We have revised the paragraph:

Lines 65:

“In those receiving nirmatrelvir/ritonavir or molnupiravir, the incidence of virologic rebound is similar to those not receiving antivirals and often accompanied by symptom rebound [2, 5-8]. Nevertheless, inconsistent findings were found in several studies reporting more frequent virologic rebound in treated individuals e.g., [4,9].”

We have also ensured that the meaning of other sentences citing this reference is correctly discussed.